# MMGenBench: Fully Automatically Evaluating LMMs from the Text-to-Image Generation Perspective

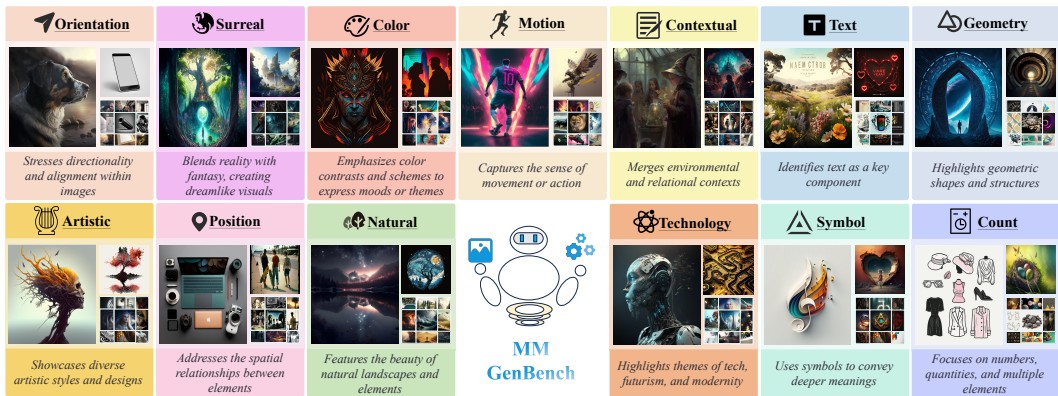

Figure 1: The MMGenBench-Test consists of 13 distinct image patterns, each of which includes several images. The text, accompanied by a corresponding pattern, serves as a concise explanation of that specific image pattern. Please refer to the Appendix B.1 for more details.

## Abstract

Large Multimodal Models (LMMs) demonstrate impressive capabilities. However, current benchmarks predominantly focus on image comprehension in specific domains, and these benchmarks are labor-intensive to construct. Moreover, their answers tend to be brief, making it difficult to assess the ability of LMMs to generate detailed descriptions of images. To address these limitations, we propose the MMGenBench-Pipeline, a straightforward and fully automated evaluation pipeline. This involves generating textual descriptions from input images, using these descriptions to create auxiliary images via text-to-image generative models, and then comparing the original and generated images. Furthermore, to ensure the effectiveness of MMGenBench-Pipeline, we design MMGenBench-Test, evaluating LMMs across 13 distinct image patterns, and MMGenBench-Domain, focusing on generative image performance. A thorough evaluation involving over 50 popular LMMs demonstrates the effectiveness and reliability of both the pipeline and benchmark. Our observations indicate that numerous LMMs excelling in existing benchmarks fail to adequately complete the basic tasks related to image understanding and description. This finding highlights the substantial potential for performance improvement in current LMMs and suggests avenues for future model optimization. Concurrently, MMGenBench-Pipeline can efficiently assess the performance of LMMs across diverse domains using only image inputs. All code and data will be released.

## 1 Introduction

We have witnessed rapid progress of Large Multimodal Models (LMMs) Huang et al. (2023); Wang et al. (2023); Lu et al. (2024b); Chu et al. (2024); Wang et al. (2024); Chen et al. (2023); OpenAI (2024), which efficiently utilize the strength of LLMs OpenAI (2023); Team et al. (2023); OpenAI (2024); Wu et al. (2023); Kasneci et al. (2023); Meta (2024) in processing visual and textual inputs.

Figure 2: Comparison between previous benchmarks and MMGenBench. MMGenBench has several novel features: 1) Based on powerful text-to-image models and image representation models, MMGenBench can fully automatically complete the evaluation of LMMs without the need for expensive manual annotation; 2) MMGenBench can easily evaluate the performance of LMMs in any domain, whereas previous benchmarks could mostly only evaluate the performance in specific domains; 3) The "answer" to previous benchmarks were mostly brief, overlooking the basic ability to generate detailed descriptions of images.

Compared to text, visual images are characterized by their high level of abstraction and information density, while exhibiting strong spatial correlations and structural complexity. The development of comprehensive benchmarks is crucial for enhancing and accurately evaluating LMMs. Specifically, many popular benchmarks Lin et al. (2014); Hudson & Manning (2019); Yin et al. (2024); Xu et al. (2023); Zhang et al. (2024); Fu et al. (2023; 2024b); Liu et al. (2024c); Ouyang et al. (2024), provide standardized evaluations for LMMs by assessing multimodal tasks across various datasets. However, these benchmarks frequently depend on traditional datasets, resulting in issues of data leakage and limited task diversity. Although many various tasks such as VQA Lu et al. (2024a); Yu et al. (2023b); Ma et al. (2024); Liu et al. (2024d); Ouyang et al. (2024), Image Caption Ke et al. (2019); Liu et al. (2024b), and OCR Liu et al. (2024c); Fu et al. (2024a); Yang et al. (2024) are carefully designed, as shown in Fig. 2(a), most existing benchmarks require expensive manual construction and focus on specific domains, making it difficult to extend to other domains. The constructed sample answers are mostly brief, focusing only on evaluating the understanding performance of LMMs and ignoring the evaluation of the ability to generate detailed descriptions of images.

In addition, the understanding and generation of images remain disparate fields, with the most powerful models Esser et al. (2024); Labs (2024); OpenAI (2024) in their respective domains adhering to distinct paradigms. For instance, GPT-4 OpenAI (2024), which is grounded in the next token prediction paradigm, exhibits an impressive capacity for image comprehension. Similarly, Flux Labs (2024) has achieved noteworthy success in text-to-image synthesis by leveraging diffusion models Ho et al. (2020); Rombach et al. (2022). This divergence highlights the complexity of achieving a unified approach to image processing and synthesis, as the state-of-the-art techniques continue to evolve along separate trajectories. Furthermore, LMMs are extensively employed to generate data for generative models Pei et al. (2024); Wu et al. (2024b); Lian et al. (2024); Huang et al. (2024a). It is noteworthy that LMMs excel in image-to-text tasks, while diffusion models are particularly effective in text-to-image tasks. A robust understanding of an image implies that LMMs can distill its essential information into text prompts, which text-to-image models can use to reconstruct the scene to a certain extent. This process can be viewed as a form of "compression". Hence, it is both reasonable and significant to evaluate the performance of LMMs using diffusion models.

In this paper, we propose MMGenBench-Pipeline, a fully automated evaluation pipeline (refer to Fig. 2(b)) that initially allows LMMs to generate textual descriptions from input images, then employs text-to-image generative models to create auxiliary images. Finally, we use an image representation model to obtain the embeddings of images and perform post-processing to assess the performance of LMMs in image understanding and description. To verify the effectiveness of MMGenBench-Pipeline, we introduce MMGenBench-Test, a comprehensive benchmark designed to evaluate LMMs across 13 distinct image patterns (refer to Fig. 1), and MMGenBench-Domain, which focuses on assessing LMMs performance within the generative image domain. Extensive experiments on over 50 popular LMMs demonstrate the effectiveness and reliability of both the pipeline and benchmark. Notably, our findings reveal that numerous LMMs excelling in existing benchmarks fail to address the basic tasks of image understanding and description, highlighting the substantial potential for improvement and optimization in future models. The MMGenBench-

Pipeline also enables efficient evaluation of LMMs across diverse domains through image inputs alone, providing a flexible and scalable benchmarking tool.

In summary, our contributions are as follows:

- **Fully Automated Evaluation Pipeline**: We propose the first fully automated pipeline MMGenBench-Pipeline designed to evaluate the capabilities of LMMs in image understanding and description by solely utilizing images. This pipeline utilizes text-to-image models and image representation models for automated evaluation, thereby markedly minimizing human involvement and improving the efficiency and objectivity of the evaluation procedure.

- **Comprehensive Benchmarks**: In order to verify the effectiveness of MMGenBench-Pipeline, we developed MMGenBench-Test, a comprehensive benchmark designed to evaluate LMMs across 13 image patterns, and MMGenBench-Domain, which assesses the performance of LMMs in the generative image domain.

- **Extensive Evaluation**: Our study includes a broad evaluation of over 50 popular LMMs, providing critical insights into their capabilities and limitations in basic image understanding and description tasks.

Please refer to Appendix A for related work.

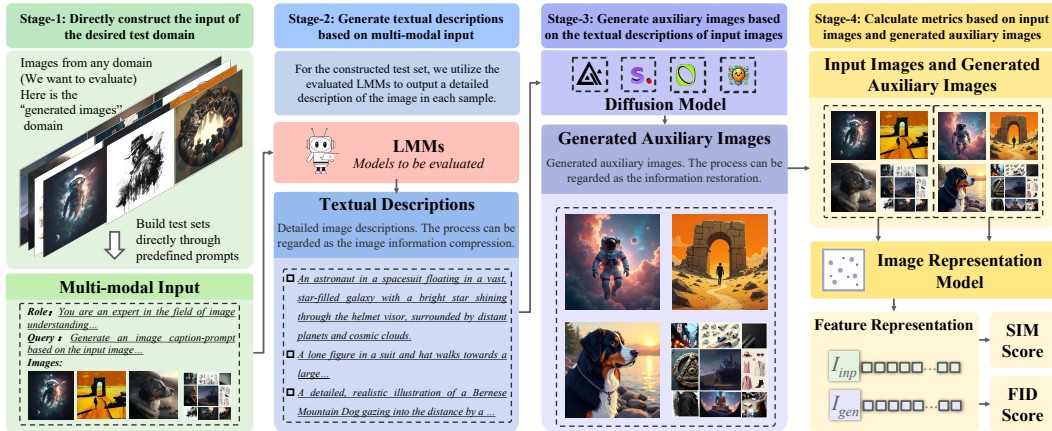

Figure 3: An overview of the MMGenBench-pipeline, illustrating the fully automated evaluation process. It starts by receiving user input (including the task instruction prompt and input images), and then generates the corresponding textual descriptions of input images. Subsequently, this process is followed by using a powerful text-to-image model to generate auxiliary images, then produces the representation of the input images and the generated ones using an image representation model, and finally outputs the evaluation score of LMMs.

## 2 THE MMGENBENCH-PIPELINE

### 2.1 FULLY AUTOMATED EVALUATION PIPELINE

The proposed pipeline for LMMs, based on text-to-image generative models and image representation models, consists of four components: test set construction, textual description generation, auxiliary image generation, and quantitative metric computation.

**Test Set Construction.** Our method can easily evaluate image understanding and description capabilities of LMMs in any domain. As shown in Stage 1 of Fig. 3, given any image from the domain to be tested, we can construct the multi-modal input for the LMMs to be evaluated simply using the predefined prompts, and apply it to the subsequent process.

**Textual Description Generation.** The input modalities for the LMM primarily consist of two parts: the image used for model understanding and inference, and the prompt guiding the inference

direction. To facilitate a standardized workflow, we employ a manually crafted, normalized prompt $\mathbf{P}_{art}$. This prompt constrains the LMM's comprehension and reasoning across five dimensions: role, definition explanation, task instruction, key points and requirements, and output format (see Appendix C.1 for details). This process can be formalized as follows:

$$\mathbf{P}_{gen} = \text{LMM}(\mathbf{I}_{inp}, \mathbf{P}_{art}). \tag{1}$$

Through the above process, we obtain the detailed textual description of an image $\mathbf{P}_{gen}$, generated by the LMM, which will serve as the input for the subsequent stage.

**Auxiliary Image Generation.** In this stage, the main process involves the utilization of text-to-image models to generate auxiliary images based on the given textual descriptions. Theoretically, the generation quality is highly dependent on the chosen text-to-image model. To minimize variable interference, we standardize the evaluation by selecting four state-of-the-art models: FLUX.1-dev Labs (2024), Stable Diffusion 3.5 Esser et al. (2024), Kolors Team (2024) and Lumina Gao et al. (2024a). The comparison results across these models provide cross-validation of generation effectiveness. This phase is represented as:

$$\mathbf{I}_{gen} = G(\epsilon; \mathbf{P}_{gen}, \theta), \tag{2}$$

where $\mathbf{I}_{gen}$ denotes the generated image, $\epsilon$ represents a randomly sampled variable from the latent space, typically following a certain distribution (e.g., Gaussian distribution), $\theta$ represents the model parameters, and $G$ denotes the image generation function.

**Quantitative Metric Computation.** We quantify the functionality of LMM by evaluating the similarity between the input image $\mathbf{I}_{inp}$ and the generated one $\mathbf{I}_{gen}$. Since most generative models introduce a certain degree of randomness in their inference process, achieving pixel-level consistency between images is challenging. To address this, we conduct comparisons at the representational level. Specifically, we utilize the Unicom An et al. (2023) model to encode each image and obtain its representations. This phase is encapsulated as:

$$\mathbf{F}_{I_{inp}} = \text{Encoder}(\mathbf{I}_{inp}), \mathbf{F}_{I_{gen}} = \text{Encoder}(\mathbf{I}_{gen}), \tag{3}$$

where $\mathbf{F}_{I_{inp}}$ and $\mathbf{F}_{I_{gen}}$ represent the features extracted from the input image $\mathbf{I}_{inp}$ and the generated image $\mathbf{I}_{gen}$, respectively. To provide a quantitative evaluation, we compute both a similarity score and a generation quality score based on these feature representations.

## 2.2 EVALUATION METRIC

SIM-Score is a metric for evaluating the similarity between two features, commonly using cosine similarity. In this study, SIM-Score is calculated as follows:

$$\text{SIM-Score}(\mathbf{I}_{inp}, \mathbf{I}_{gen}) = \frac{\mathbf{F}_{I_{inp}} \cdot \mathbf{F}_{I_{gen}}}{\|\mathbf{F}_{I_{inp}}\|\|\mathbf{F}_{I_{gen}}\|}, \tag{4}$$

where $\cdot$ denotes the dot product operation, and $\|\cdot\|$ represents the vector norm, used for normalization. The SIM-Score ranges between $-1$ and $1$, where a value of $1$ indicates maximum similarity, $0$ denotes no similarity, and $-1$ signifies complete opposition.

The FID Score (FID-Score) measures the difference between the distributions of generated images and real images. A lower FID Score indicates that the generated images have higher quality and greater similarity to the real images. The calculation is primarily based on the below equation:

$$\text{FID} = \|\mu_x - \mu_y\|^2 + \text{Tr}(\Sigma_x + \Sigma_y - 2(\Sigma_x \Sigma_y)^{1/2}), \tag{5}$$

where $\mu_x$ and $\mu_y$ denote the means of the input image features $F_{I_{inp}}$ and the generated image features $F_{I_{gen}}$, respectively, while $\Sigma_x$ and $\Sigma_y$ represent their covariances. The notation $\text{Tr}(\cdot)$ denotes the trace of a matrix.

## 3 THE MMGENBENCH BENCHMARK

### 3.1 OVERVIEW

Although the existing benchmarks can evaluate the image understanding capabilities of LMMs from various dimensions, as mentioned in Sec. 1, they still lack the evaluation of the basic image understanding of LMMs and the ability to describe the images clearly.

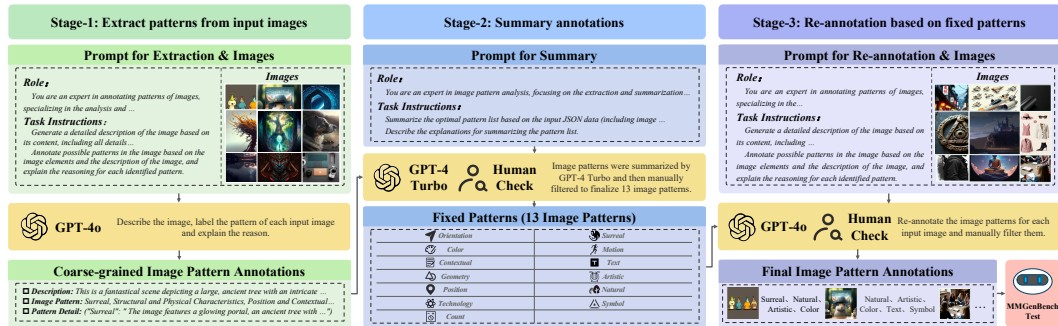

Figure 4: An overview of the MMGenBench-Test benchmark construction process. We first use GPT-4o to extract the image patterns from the input images. Then, we use GPT-4 Turbo to summarize these patterns and manually select 13 patterns. Subsequently, GPT-4o is employed again to re-annotate these patterns. These annotations are reviewed and modified to produce the final result by human annotators.

To effectively measure the understanding and description capabilities of LMMs across various types of images, we constructed a high-quality test set MMGenBench-Test for 13 image patterns using the JourneyDB Sun et al. (2024) test set. We proposed a multi-stage method for extracting and annotating image patterns as illustrated in Fig. 4. To ensure the results' accuracy, we manually double-checked the image patterns and performed the final annotations.

In addition, we have also constructed a dataset in the "image generation" domain, termed MMGenBench-Domain, to evaluate the ability of LMMs in the understanding and describing "generated images" task. It is important to emphasize that MMGenBench-Pipeline can measure the ability of LMMs to understand and generate detailed image descriptions in any domain. By utilizing images from a particular domain, we can easily assess the performance of LMMs specific to that domain.

## 3.2 DATASET STATISTICS

In the MMGenBench-Test dataset, we constructed a high-quality test set containing $1,284$ images across 13 distinct image patterns (see Fig. 1). The distribution of images per pattern is shown in Fig. 5, which illustrates that each pattern contains a minimum of $114$ images. Please note that an image may contain multiple patterns. For instance, the first image annotation in Fig. 4 contains four image patterns: "Surreal", "Natural", "Artistic" and "Color". These 13 image patterns are carefully designed so that the dataset can measure the image comprehension and description capabilities of LMMs across diverse dimensions. Additionally, all samples in the dataset are manually checked and annotated to ensure their overall quality.

To construct MMGenBench-Domain, we randomly sampled $10,000$ images from the JourneyDB validation set. By utilizing the MMGenBench-pipeline, we can evaluate the image understanding and descriptive performance of LMMs within this domain, obviating the need for additional data.

## 3.3 DATA COLLECTION

To construct MMGenBench-Test, we first extract all images from the JourneyDB test set and process them using the process shown in Fig. 4. For the domain-specific dataset, we use the JourneyDB validation set for its construction. Subsequently, we will elaborate on each processing step.

**Pattern Extraction.** We extract image patterns from existing images to measure the understanding and description ability of LMMs across various image categories. Given that we possess only images but no additional information, we utilize the powerful GPT-4o model to extract and analyze the underlying image patterns. The task is executed by GPT-4o in three sequential steps: 1) providing a detailed description of the image; 2) annotating the possible patterns based on the image features and description; and 3) explaining the rationale behind the annotated image patterns. By utilizing our carefully designed prompts (refer to Appendix B.3), GPT-4o can perform the task effectively. An example is presented in Fig. 10 in Appendix B.2, wherein the image description, alongside the

extracted image patterns and rationales, aligns accurately with the image content. By annotating each image in the JourneyDB test set, we ultimately obtained a total of 1868 image patterns.

**Annotation Summary.** By leveraging GPT-4o, we extracted an extensive range of image patterns and observed that the semantic meanings across different patterns may be consistent. We conducted thorough statistics of the image patterns, by quantifying the occurrences of each type and ranking them in descending order (as seen in Appendix B.1). Subsequently, GPT-4-Turbo was utilized to generate the summary. To ensure the accuracy and validity of the extracted patterns, we carefully designed model prompts for this task (shown in Appendix B.3). This procedure ultimately resulted in a hand-crafted list of 13 distinct image patterns, each thoroughly described in Appendix B.1.

**Image Re-annotation.** We conducted a re-annotation of the JourneyDB test set using the previously identified 13 image patterns. Similar to Image Pattern Extraction, GPT-4o was employed to generate descriptions for each image. Subsequently, these descriptions were annotated based on the predefined image patterns, accompanied by the underlying reasons. A key difference in this process is that re-annotation is restricted to the specified 13 image patterns, excluding any other patterns. Given the extensive number of images in the test set, it is essential to select a subset of the images for subsequent operations. The re-annotated images were systematically classified into distinct image patterns, each containing a specified number of images. To construct the final dataset, we randomly sampled a subset of images from each pattern with a selection probability of $\frac{100}{N}$, where $N$ represents the total number of images within the pattern. To mitigate potential annotation errors, we manually verified and annotated each image. The final dataset was established by voting between the model predictions and annotations.

**Domain Data Collection.** Constructing high-quality datasets is an extremely expensive task. Therefore, it is crucial to develop an efficient and convenient method for creating domain-specific datasets when evaluating the image understanding and description capabilities of LMMs in a new domain. By leveraging the MMGenBench-pipeline, our method enables a seamless evaluation of LMM performance across various domains, only depending on the availability of domain-specific images. In this study, we randomly select $10,000$ images from the JourneyDB validation set to create our domain-specific dataset. To quantify LMM performance within this domain, we calculate both FID-Score and SIM-Score.

## 4 EXPERIMENT

### 4.1 EXPERIMENTAL SETUP

**Evaluation Models.** In the evaluation of LMMs, we selected over 50 models that exhibited strong performance on the VLM leaderboard Duan et al. (2024). We mainly focus on open-source models, which include Qwen2-VL Wang et al. (2024), InternVL2 Chen et al. (2023), LLaVA-OV Li et al. (2024a), Ovis Lu et al. (2024b), etc. Additionally, we evaluated three closed-source API models: GPT-4o OpenAI (2024), Qwen-VL-Max Bai et al. (2023), and Qwen-VL-Plus Bai et al. (2023). To process the textual descriptions generated by LMMs and create auxiliary images, we selected four text-to-image generation models: FLUX.1-dev Labs (2024), Stable Diffusion 3.5 Esser et al. (2024), Kolors Team (2024) and Lumina Gao et al. (2024a). During the final evaluation phase, the Unicom An et al. (2023) image representation model was utilized to extract image features. Further details about our baseline models are provided in the Appendix C.2.

**Implementation Details.** During the textual descriptions generation process in LMMs, we utilize the default settings of VLMEvalKit Duan et al. (2024), modifying only the predefined query prompt and the image to meet the task-specific requirements. Subsequently, we employ four distinct text-to-image models to generate auxiliary images and extract features using an image representation model. We then compute both the SIM-Score and FID-Score for evaluation. We analyzed the performance metrics of four text-to-image models on the MMGenBench-Test dataset, as illustrated in Fig. 6. The results indicate that the four text-to-image models exhibited consistency in both SIM-Score and FID-Score. Therefore, unless otherwise stated, the reported results are obtained using the FLUX.1-dev text-to-image model. Please refer to Appendix C.2 for more detailed results.

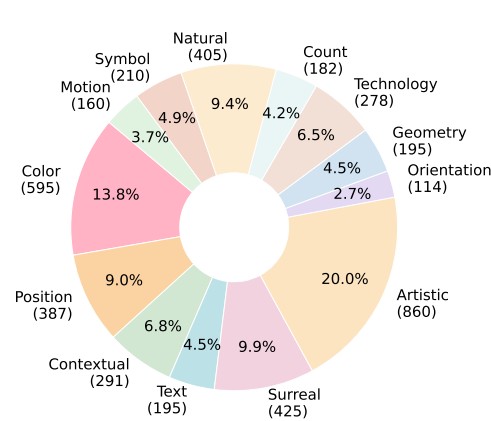

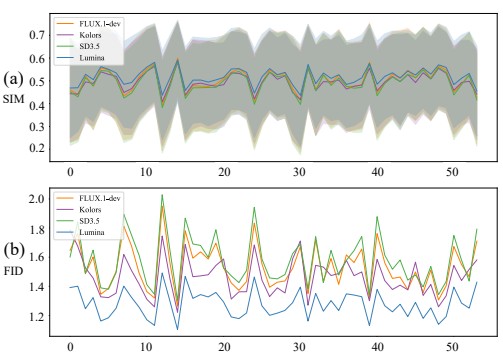

Figure 5: Statistics of MMGenBench-Test, which contains 13 image patterns with 1,284 images. More details are in Sec. 3.

Figure 6: The comparative analysis of four different text-to-image models on MMGenBench. The horizontal axis denotes the index of LMMs. Fig. (a) illustrates the SIM-Score for over 50 LMMs on MMGenBench-Test, while Fig. (b) presents the FID-Score for the same set of models on MMGenBench-Test.

| Model | Test | | Domain | |
|---|---|---|---|---|
| | SIM↑ | FID↓ | SIM↑ | FID↓ |
| GPT-4o OpenAI (2024) | 0.566 | 1.306 | - | - |
| Qwen-VL-Max Bai et al. (2023) | 0.552 | 1.363 | - | - |
| Qwen-VL-Plus Bai et al. (2023) | 0.475 | 1.586 | - | - |
| Qwen2-VL-72B Wang et al. (2024) | 0.553 | 1.357 | 0.545 | 0.710 |
| Qwen2-VL-7B Wang et al. (2024) | 0.532 | 1.437 | 0.524 | 0.775 |
| Qwen2-VL-2B Wang et al. (2024) | 0.487 | 1.549 | 0.501 | 0.806 |
| InternVL2-76B Chen et al. (2023) | **0.599** | **1.264** | **0.599** | **0.632** |
| InternVL2-40B Chen et al. (2023) | 0.566 | 1.350 | 0.566 | 0.696 |
| InternVL2-26B Chen et al. (2023) | 0.576 | 1.320 | 0.577 | 0.671 |
| InternVL2-8B Chen et al. (2023) | 0.547 | 1.403 | 0.548 | 0.701 |
| InternVL2-4B Chen et al. (2023) | 0.556 | 1.345 | 0.556 | 0.689 |
| InternVL2-2B Chen et al. (2023) | 0.476 | 1.563 | 0.483 | 0.848 |
| Ovis1.6-Gemma2-9B Lu et al. (2024b) | 0.582 | 1.316 | 0.579 | 0.667 |
| Ovis1.5-Gemma2-9B Lu et al. (2024b) | 0.521 | 1.487 | 0.524 | 0.808 |
| Ovis1.5-Llama3-8B Lu et al. (2024b) | 0.526 | 1.466 | 0.527 | 0.795 |
| LLaVA-OV-72B Li et al. (2024a) | 0.494 | 1.561 | 0.491 | 0.872 |
| LLaVA-OV-SI-72B Li et al. (2024a) | 0.512 | 1.512 | 0.514 | 0.813 |
| LLaVA-OV-7B Li et al. (2024a) | 0.488 | 1.590 | 0.490 | 0.861 |
| LLaVA-OV-SI-7B Li et al. (2024a) | 0.492 | 1.554 | 0.497 | 0.834 |
| MiniCPM-V2.6-8B Yao et al. (2024) | 0.548 | 1.386 | 0.545 | 0.710 |
| RBDash-72B RBDash-Team (2024) | 0.525 | 1.413 | 0.527 | 0.740 |
| xGen-MM-4.4B Xue et al. (2024) | 0.414 | 1.671 | 0.412 | 0.940 |
| MiniCPM-V2.5-8B Yao et al. (2024) | 0.526 | 1.432 | 0.530 | 0.767 |

Table 1: Experiment results of different LMMs on MMGenBench-Test/Domain, where SIM corresponds to SIM-Score, FID corresponds to FID-Score. More details are in Sec. C.3.

## 4.2 MAIN RESULTS

**Overall Performance.** As shown in Table 1, the SIM-Score of the most advanced LMMs on MMGenBench-Test is below 0.600. Specifically, GPT-4o achieved a score of 0.566, which is lower than the 0.599 obtained by the open-source model InternVL2-76B. Notably, there is no clear correlation between the model size and performance across different series of models. This indicates the importance of training data and the training process in developing LMMs with strong capabilities of image understanding and description. Furthermore, it is observed that models excelling on existing benchmarks do not necessarily perform well on MMGenBench-Test, such as LLaVA-OV, which

only scores $0.494$. In the MMGenBench-Domain dataset, the SIM-Score results align closely with those of the MMGenBench-Test. However, there is a significant difference in FID-Score because MMGenBench-Domain includes 10,000 images, thereby improving the accuracy of its FID-Score measurement. Therefore, we propose using SIM-Score as the primary metric. Detailed results and comprehensive analysis are provided in the following sections as well as in Appendix C.3-C.9.

**Pipeline Effectiveness.** To verify the pipeline, we conducted a user study. We randomly selected 1850 samples from MMGenBench-Test. Each sample contains an original image and two outputs from different LMMs, which were randomly selected from 6 mainstream LMMs. We determined which LMM's output quality was better or if both were comparable through human annotation. Then we calculate the SIM score. In cases where the difference in SIM scores was less than $0.01$, we considered comparable; otherwise, we deemed the LMM with the higher score to have better output. We collect the human metric based on votes from 9 human experts. The results showed that in $88.27\%$ of cases, our pipeline was well aligned with human. More details are in Appendix. C.4.

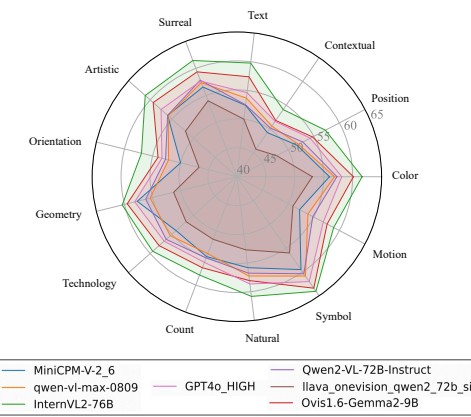

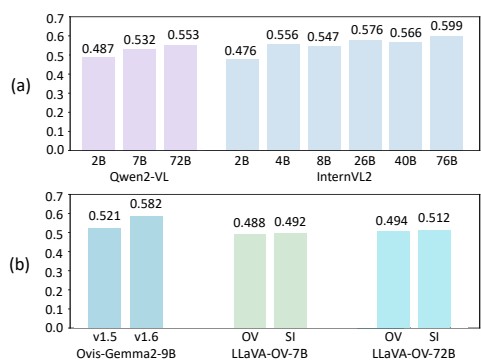

Figure 7: Model performance by image patterns. Please refer to Sec. 4.2 for more results and discussions.

Figure 8: The comparison of SIM-Score with different LMMs on MMGenBench-Test: (a) LMMs from the same series but with different parameter sizes; (b) LMMs with the same sizes but using different training protocol or data. More details are in Sec. 4.2.

**Image Patterns Revealing Model's Strengths and Limitations.** Fig. 7 shows the accuracy of the best-performing LMMs on the MMGenBench dataset. We observed that LMMs exhibit superior performance on image patterns such as "Artistic", "Surreal", "Symbol", "Color", etc. Conversely, their performance declines on patterns such as "Contextual", "Orientation", "Count", "Motion", etc. This discrepancy suggests that LMMs are proficient in tasks requiring coarse-grained image understanding and description. In contrast, their abilities for fine-grained understanding and description are more challenging, as these tasks require the understanding of the complex contextual relationships within images and the precise linguistic description.

**Impact of Parameter Scales, Training Methods, and Data Variations.** Fig.8(a) shows the performance results of the same series of models with varying parameter sizes, while Fig.8(b) presents the performance results of the same model and parameters under different training protocols and datasets. We can see that as the model parameters increase, the performance of Qwen2-VL improves from $0.487$ to $0.553$, and the results of InternVL2 increased from $0.476$ to $0.599$. Additionally, by using improved training data and protocols, Ovis1.6 outperforms Ovis1.5 by $0.06$. Given that our evaluation is based on a single input image, LLaVA-OV-SI demonstrates superior performance compared to LLaVA-OV.

### 4.3 ANALYSIS

We identify three common problems that significantly affect the performance of LMMs in image understanding and description. Please refer to the Appendix D for more details.

**Failure in Following Instructions.** As shown in Fig. 9, we have selected various LMMs to demonstrate a portion of the qualitative examples. It is observed that certain LMMs fail to strictly follow instructions when generating the textual descriptions. Specifically, some LMMs prepend a prefix to

Figure 9: Qualitative Results on MMGenBench. We present common problems identified in the experiments, which include issues with output format, generated content, and final results. Please refer to Sec. 4.3 and Appendix D for more discussions and results.

the textual descriptions (e.g., RBDash-72B RBDash-Team (2024) in Fig. 9), while others append explanatory content following the textual descriptions (e.g., MiniCPM-Llama3-V2.5 Yao et al. (2024) in Fig. 9). Additionally, it is also evident that the ability to follow instructions does not appear to be correlated with the parameter size of different series LMMs. For example, both Qwen2-VL-2B and InternVL2-2B accurately generate the textual descriptions following the given instructions. We argue that an effective LMM, especially when undergoing instruction tuning, should be capable of accurately following instructions and successfully completing the task.

**Inability in Generating Detailed Textual Descriptions.** Most of the image-text pairs used in the training of existing LMMs have relatively short image captions. Consequently, numerous models exhibit suboptimal performance on the MMGenBench Benchmark, which requires the model to understand the image and provide a detailed description. As shown in Fig. 9, LLaVA-OV-72B Li et al. (2024a), despite its extensive 72B parameters, produces relatively short textual descriptions, which hinders its ability to excel on the MMGenBench Benchmark. Additionally, we observed that for LMMs in the same series, smaller LMMs tend to generate shorter textual descriptions, as evidenced in the Qwen2-VL and InternVL2 series. Based on these findings, we hypothesize that incorporating image-description pairs during the training phase of LMMs could achieve superior results.

**Model Overfitting.** Existing LMMs frequently undergo task-specific training to meet established benchmarks. This specialized training can lead to overfitting, which may compromise the basic image understanding and description capabilities of LMMs. For instance, the xGen-MM Xue et al. (2024) results in Fig. 9 demonstrate an overfitting towards the notion of "safety", despite the given image not presenting any safety risk. This observation suggests that in adapting models to various tasks, it is crucial to maintain the basic image understanding and description capabilities of LMMs.

## 5 CONCLUSION

In this paper, we present a straightforward and fully automated evaluation pipeline to comprehensively evaluate the ability of LMMs in image understanding and description. Based on MMGenBench-pipeline, we introduce MMGenBench-Test and MMGenBench-Domain to evaluate LMM performance across different image patterns and domain-specific images. By evaluating over 50 widely used LMMs, we demonstrate the reliability and effectiveness of both the pipeline and benchmarks. In conclusion, our method provides a more fundamental evaluation of existing LMMs, identifies specific gaps in their image understanding and description capabilities, and establishes a robust foundation for further research and model improvement in these areas. All code and data will be released soon.

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

## A    RELATED WORK

### A.1    AUTOMATIC BENCHMARKS

The rapid advancements in LLMs have prompted the development of diverse benchmarks to evaluate their performance. Early efforts, such as LMExamQA Bai et al. (2024), introduced the "Language-Model-as-an-Examiner" framework to provide a scalable and comprehensive evaluation for LLMs. However, there remains a significant gap in the evaluation of the reasoning capabilities of LLMs, especially in dynamic contexts. To address this deficiency, recent frameworks like DYVAL Zhu et al. (2023) and DYVAL2 Zhu et al. (2024) have focused on dynamic reasoning tasks. DYVAL concentrates on reasoning abilities, while DYVAL2 extends the evaluation to a broader psychometric approach, thereby providing deeper insights into cognitive evaluations. Additionally, AutoBencher Li et al. (2024b) has automated the generation of challenging and novel datasets, specifically designed for the evaluation of LLM. Platforms such as UNIGEN Wu et al. (2024a) and Task Me Anything Zhang et al. (2024) aim to further enhance evaluation by developing domain-specific benchmarks tailored to the unique capabilities of LLMs.

### A.2    LMM BENCHMARKS

The emergence of LMMs has highlighted the insufficiency of traditional benchmarks, which primarily focus on isolated task performance and fail to capture the full capabilities exhibited by multimodal models. Early multimodal benchmarks, such as those introduced by Lin et al. (2014); Goyal et al. (2017); Yu et al. (2023b), established essential foundations but lacked the granularity required for robust evaluation of the model's understanding and reasoning abilities. Recent studies Liu et al. (2025); Chen et al. (2024a); Yu et al. (2023a) emphasize the critical need for comprehensive, fine-grained benchmarks that can more accurately evaluate the complex capabilities of LMMs. However, existing benchmarks, such as LVLM-eHub Xu et al. (2023) and LAMM Yin et al. (2024), still rely on classical datasets that do not fully reflect the current state of the field, and neglect issues such as data leakage during model training. Addressing these concerns, MMStar Chen et al. (2024a) introduces an innovative approach that eliminates visual content dependencies and mitigates data leakage risks, thereby providing a more reliable and secure evaluation framework. Furthermore, the development of automated benchmarking systems, such as AUTOBENCH-VBao et al. (2024), enhances the scalability and objectivity of the evaluations, by automating the benchmarking process and minimizing human bias. Despite these significant advancements, further refinement is needed to create truly comprehensive and dynamic benchmarks that can capture the evolving capabilities of LMMs.

## B    DETAILS OF MMGENBENCH BENCHMARK

### B.1    DETAILS OF MMGENBENCH-TEST DATA

**Thirteen Image Patterns.** As illustrated in Fig. 11, summarization is performed using GPT-4 Turbo, followed by human verification. We identify 13 distinct image patterns, each with a comprehensive explanation. It should be noted that a single image can exhibit multiple patterns concurrently.

**Patterns Extracted by GPT-4o.** Fig. 12 shows the image patterns extracted by GPT-4o, organized in descending order. While the total number of extracted patterns is 1868, only the most frequently occurring patterns are presented due to space constraints.

### B.2    CASE STUDY OF PATTERN EXTRACTION

GPT-4o has demonstrated strong capabilities in extracting image patterns from given images. Fig. 10 illustrates an instance where GPT-4o effectively identifies and extracts patterns from an image. The results demonstrate a high degree of correspondence between the image description, the extracted pattern, and the corresponding explanation, all of which align precisely with the visual content.

### B.3 DETAILS OF CONSTRUCTION PROMPTS

To construct the MMGenBench-Test, a series of prompts have been tailored to the specific requirements of our task meticulously. Fig. 13, 14, and 15 include the three sequential stages in the pipeline (see Fig. 3) used to develop MMGenBench-Test.

## C ADDITIONAL EXPERIMENTAL DETAILS

### C.1 EVALUATION PIPELINE PROMPT

We meticulously crafted a prompt to guide LMMs in generating textual descriptions of images for subsequent evaluation. As shown in Fig. 16, the prompt encompasses five key perspectives: "role", "definition", "task", "key points", and "output". To ensure the objectivity and consistency of the evaluation, we employed this singular prompt, as it is sufficiently clear to enable the model to effectively perform the textual description generation task.

### C.2 EXPERIMENTAL SETUP

We employed over 50 popular LMMs, as detailed in Tables 7 and 8, which are available on the OpenVLM Leaderboard Contributors (2023). For open-source LMMs, we utilized the original configuration of VLMEvalKit Duan et al. (2024) to perform inference of textual descriptions of images on 8 NVIDIA-H20 GPUs. During the image generation and metric calculation stages, we still use 8 NVIDIA-H20 GPUs for multi-process inference to efficiently complete the task.

### C.3 COMPREHENSIVE EXPERIMENTAL EVALUATION

A comprehensive set of experimental results encompassing over 50 popular LMMs is shown in Tables 7 and 8.

### C.4 HUMAN ALIGNMENT SCORE

Below we describe the complete experimental setup, annotator protocol, and statistical results.

**User Study Setup.** We select 6 representative LMMs: GPT-4o-high, InternVL2-76B, Qwen2-VL-72B-Instruct, LLaVA-OneVision-Qwen2-72B, MiniCPM-V-2.6, and Ovis1.6-Gemma2-9B. We construct 1,000 paired caption comparisons, where each comparison is formed by: (1) randomly sampling two models from the six, (2) randomly selecting one image from MMGenBench-Test, and (3) presenting the two model-generated captions to human annotators.

**Annotator Task.** Each of the 1,000 examples is independently labeled by 9 human annotators. For each example, annotators are instructed to choose:

- **C1** if caption-1 describes the image better,
- **C2** if caption-2 describes the image better, or
- **B** if the two captions are comparable.

For example, an annotation list

$$['C1','C1','C1','C1','C1','C1','C1','C2','B']$$

indicates that 7 annotators prefer caption-1, 1 prefers caption-2, and 1 considers them similar.

**Agreement Computation.** Our pipeline predicts the outcome using the SIM-score difference:

- $\Delta > 0.01 \Rightarrow$ **C1**,
- $\Delta < -0.01 \Rightarrow$ **C2**,
- $|\Delta| \leq 0.01 \Rightarrow$ **B**.

We compute agreement against all $9 \times 1000 = 9,000$ individual human votes. The pipeline achieves strong alignment with human judgments:

- Agreement with individual human votes: $7944/9000 = 88.27\%$,
- Agreement with human majority decisions ($\geq 5/9$ votes): $900/1000 = 90.0\%$.

**Pattern-Wise Agreement.** To investigate whether certain patterns are inherently more ambiguous or less consistent, we further compute vote agreement separately for patterns with sufficiently large numbers of human-evaluated samples (approximately $\geq 100$), ensuring statistical reliability.

| Pattern | Right | Wrong | Agreement (%) |
|---|---|---|---|
| Natural | 216 | 23 | 90.4 |
| Symbol | 159 | 17 | 90.3 |
| Technology | 125 | 15 | 89.3 |
| Artistic | 111 | 16 | 87.4 |
| Count | 99 | 10 | 90.8 |
| Motion | 109 | 9 | 92.4 |

Table 2: Pattern-wise agreement between MMGenBench and human judgment.

**Key Takeaway.** Across all evaluated patterns, agreement between the proposed metric and human judgment consistently falls within 87%-92%. These patterns cover the majority of MMGenBench-Test and represent diverse visual skills, including object fidelity (Natural), symbolic semantics (Symbol), dynamic information (Motion), and fine-grained detail (Count). Overall, the results demonstrate that the pipeline's alignment with human perception is both high and stable across diverse image types.

## C.5 SENSITIVITY ANALYSIS OF T2I MODELS

We conducted a comprehensive sensitivity analysis on four different T2I models (Flux-1-dev, SD3.5, Kolors, Lumina). The results show:

(1) **LMM rankings are highly stable across T2I models.** SIM correlations across the four generators are extremely high (Spearman = 0.98, Kendall = 0.90 on average). This indicates that changing the T2I model has minimal effect on the relative ranking of LMMs. (Table 9, 10)

(2) **FID is more sensitive, but still maintains strong cross-model consistency.** FID correlations remain high (Spearman = 0.91, Kendall = 0.78), consistent with its known dependence on generator style distribution. (Table 11, 12)

(3) **Ranking variation is mainly due to LMM robustness rather than T2I choice.** Only a small subset of weaker LMMs exhibits noticeable variance across T2I models, while strong LMMs remain stable. This suggests that the benchmark captures true LMM semantic robustness rather than artifacts of specific T2I models. (Table 14)

(4) **MMGenBench evaluations are generator-robust.** The average variation of SIM scores across T2I models is very small (std = 0.013, range = 0.03), confirming that the benchmark is not dominated by T2I controllability. (Table 15)

Overall, these results demonstrate that the final scores are not tightly entangled with any particular T2I model, and the benchmark conclusions hold consistently across different T2I models.

## C.6 SENSITIVITY ANALYSIS OF EMBEDDING MODELS

We conducted a comprehensive sensitivity analysis across three widely used embedding models (CLIP/clip-vit-large-patch14-336, DINO/dinov2-large, UNICOM/ViT-L-14-336px). The results (Table 16-19) show that MMGenBench is highly robust to the choice of embedding model. SIM scores exhibit very high cross-model correlations (Spearman = 0.98, Kendall = 0.90), and FID scores are even more consistent (Spearman = 0.99, Kendall = 0.92). These consistently high correlations indicate that the LMM rankings are effectively unchanged across CLIP, DINO, and UNICOM, demonstrating that our evaluation is insensitive to the embedding backbone.

## C.7 REAL-WORLD DOMAIN IMAGE EVALUATION

To validate the generality of MMGenBench, we constructed a real-photo subset using 2,000 COCO images and evaluated 4 representative LMMs, results are shown in Table 3.

| Model | Test Sim | Test FID | Real-World Sim | Real-World FID |
|---|---|---|---|---|
| InternVL2-8B | 0.5478 | 1.4037 | 0.4925 | 1.4353 |
| Ovis1.6-Gemma2-9B | 0.5825 | 1.3167 | 0.5441 | 1.3189 |
| Qwen2-VL-7B-Instruct | 0.5323 | 1.4370 | 0.4984 | 1.3946 |
| LLaVA-OneVision-Qwen2-7B | 0.4888 | 1.5904 | 0.4344 | 1.7266 |

Table 3: Performance comparison on the MMGenBench-Test set and the Real-World Domain subset.

We further conducted a human alignment check by randomly sampling 1,000 comparisons, each labeled by three annotators, and obtained a 92.1% (2763/3000) agreement with human judgment. These results demonstrate that MMGenBench maintains robust performance on real-world photographic data, confirming that our pipeline generalizes beyond synthetic and style-rich datasets.

## C.8 LENGTH BIAS OF LMM

We conducted a controlled study on four representative LMMs (InternVL2-8B, Ovis1.6-Gemma2-9B, Qwen2-VL-7B-Instruct, LLaVA-OneVision-Qwen2-7B) under three explicit word-count constraints: $< 20$, $20 - 60$ (our default), and $> 60$ words (Table 4).

As shown in Table 5, across all models, SIM/FID improves monotonically with length (e.g., InternVL2-8B: SIM $0.4968 \rightarrow 0.5478 \rightarrow 0.5701$; Ovis1.6-Gemma2-9B: $0.5262 \rightarrow 0.5825 \rightarrow 0.5932$), confirming that very short captions ($< 20$ words) lose essential semantics. However, the relative ordering of models remains unchanged across all three length bands, and the gains for $> 60$ word outputs are modest. This indicates that the benchmark is not driven by verbosity, but rather by the amount of semantic content conveyed. Based on these observations, the 20–60 word range provides stable and comparable setting.

| Model | $< 20$ Mean | $< 20$ Std | 20-60 Mean | 20-60 Std | $> 60$ Mean | $> 60$ Std |
|---|---|---|---|---|---|---|
| InternVL2-8B | 19.17 | 4.70 | 38.87 | 12.51 | 81.81 | 83.38 |
| Ovis1.6-Gemma2-9B | 22.49 | 6.55 | 55.85 | 12.26 | 90.43 | 18.39 |
| Qwen2-VL-7B-Instruct | 15.48 | 21.06 | 27.67 | 9.08 | 68.36 | 66.71 |
| LLaVA-OneVision-Qwen2-7B | 11.09 | 3.37 | 17.35 | 6.99 | 36.75 | 24.88 |

Table 4: Word-count statistics under different length constraints.

| Model | $< 20$ SIM | $< 20$ FID | 20–60 SIM | 20–60 FID | $> 60$ SIM | $> 60$ FID |
|---|---|---|---|---|---|---|
| InternVL2-8B | 0.4968 | 1.5337 | 0.5478 | 1.4037 | 0.5701 | 1.3171 |
| Ovis1.6-Gemma2-9B | 0.5262 | 1.4545 | 0.5825 | 1.3167 | 0.5932 | 1.2645 |
| Qwen2-VL-7B-Instruct | 0.4911 | 1.5562 | 0.5323 | 1.4370 | 0.5602 | 1.3651 |
| LLaVA-OneVision-Qwen2-7B | 0.4494 | 1.7247 | 0.4888 | 1.5904 | 0.5245 | 1.4734 |

Table 5: SIM and FID scores under different caption length constraints.

## C.9 SEED VARIANCE OF RESULTS

To quantify variance, we conducted a five-seed evaluation on the MMGenBench-Test and report the mean and std of SIM and FID for 23 main models. Across all models, the observed stochastic variation is minimal: the std of SIM ranges from 0.0002 to 0.0038, while that of FID ranges from 0.0022 to 0.0141. Moreover, the relative ranking of the models remains highly consistent across different seeds, with Spearman correlations of $\rho = 0.9975$ for SIM and $\rho = 0.9970$ for FID.

As shown in Table 6, these results indicate that multi-run fluctuations are negligible, and the scores reported in Table 1 closely match the multi-seed averages, with a maximum difference of 0.004.

Therefore, the evaluation pipeline can be regarded as stable and robust with respect to randomness in text-to-image sampling.

| Model | SIM (Mean) | SIM (Std) | FID (Mean) | FID (Std) |
|---|---|---|---|---|
| GPT-4o | 0.5656 | 0.0014 | 1.3110 | 0.0110 |
| Qwen-VL-Max | 0.5496 | 0.0033 | 1.3763 | 0.0047 |
| Qwen-VL-Plus | 0.4759 | 0.0024 | 1.5910 | 0.0115 |
| Qwen2-VL-72B | 0.5501 | 0.0009 | 1.3687 | 0.0060 |
| Qwen2-VL-7B | 0.5297 | 0.0014 | 1.4334 | 0.0085 |
| Qwen2-VL-2B | 0.4890 | 0.0016 | 1.5399 | 0.0069 |
| InternVL2-76B | 0.5985 | 0.0015 | 1.2622 | 0.0100 |
| InternVL2-40B | 0.5645 | 0.0008 | 1.3603 | 0.0059 |
| InternVL2-26B | 0.5782 | 0.0028 | 1.3147 | 0.0100 |
| InternVL2-8B | 0.5453 | 0.0002 | 1.3849 | 0.0022 |
| InternVL2-4B | 0.5584 | 0.0008 | 1.3402 | 0.0056 |
| InternVL2-2B | 0.4763 | 0.0038 | 1.5712 | 0.0132 |
| Ovis1.6-Gemma2-9B | 0.5836 | 0.0005 | 1.3010 | 0.0092 |
| Ovis1.5-Gemma2-9B | 0.5245 | 0.0011 | 1.4883 | 0.0098 |
| Ovis1.5-Llama3-8B | 0.5263 | 0.0009 | 1.4629 | 0.0116 |
| LLaVA-OV-72B | 0.4962 | 0.0010 | 1.5620 | 0.0093 |
| LLaVA-OV-SI-72B | 0.5124 | 0.0020 | 1.5094 | 0.0118 |
| LLaVA-OV-7B | 0.4866 | 0.0019 | 1.6048 | 0.0134 |
| LLaVA-OV-SI-7B | 0.4925 | 0.0022 | 1.5622 | 0.0097 |
| MiniCPM-V2.6-8B | 0.5458 | 0.0014 | 1.3786 | 0.0039 |
| RBDash-72B | 0.5239 | 0.0019 | 1.4277 | 0.0090 |
| xGen-MM-4.4B | 0.4107 | 0.0021 | 1.6727 | 0.0141 |
| MiniCPM-V2.5-8B | 0.5276 | 0.0028 | 1.4402 | 0.0130 |

Table 6: Mean and Std of SIM and FID over five random seeds on MMGenBench-Test.

## D  QUALITATIVE EXAMPLE

Qualitative analyses for more than 50 popular LMMs are illustrated in Fig. 17-21.

## E  LIMITATIONS AND FUTURE WORK

Our MMGenBench-pipeline leverages the powerful capabilities of text-to-image and image representation models. Despite our rigorous efforts to minimize errors, the evaluation pipeline may not achieve complete accuracy for tasks, which presents significant challenges to certain text-to-image and image representation models, such as those involving table images. Our research has the potential to inform and refine future methodologies for modifying target tasks and evaluation techniques, specifically addressing challenges faced by current text-to-image and image representation models. For example, in the context of table images, LaTeX could be utilized to render content and facilitate pixel-level image matching.

In addition, the adoption of the text-to-image model and the image representation model in our MMGenBench-pipeline has led to an increase in evaluation time. Despite this, it is important to note that the MMGenBench-pipeline remains more cost-effective compared to manually constructed benchmarks. Moreover, our MMGenBench-pipeline is fully automated and eliminates the need for human intervention in the evaluation of LMM performance across various domains, making it inherently more scalable than previous benchmarks. This robust scalability enables our pipeline to seamlessly adapt to more powerful models and updated datasets. As the advancements in text-to-image models continue, the time efficiency challenges we currently encounter are expected to improve. For instance, the flux-schnell requires only four steps to generate images, which can significantly accelerate our evaluation process.

*Description*

This is a fantastical scene depicting a large, ancient tree with an intricate, glowing structure built within its trunk. The tree is surrounded by a lush forest bathed in soft, ethereal light. In the foreground, a figure in a flowing cloak stands at the edge of a serene, reflective body of water, from which a glowing circular portal emanates. The tree and its surroundings are alive with vibrant foliage and mythical details, contributing to a magical and otherworldly atmosphere.

*Image Pattern*

Surrea, Structural and Physical Characteristics, Position and Contextual Relationship

*Pattern Detail*

**Surreal：** The image features a glowing portal, an ancient tree with an integrated structure, and a magical atmosphere.
**Structural and Physical Characteristics：** There is a complex structure built within the tree, including windows and possible habitation areas.
**Position and Contextual Relationship：：** The figure is positioned in the foreground, directly facing the glowing portal, adding a sense of focus and interaction with the scene.

Figure 10: A case study of image pattern extraction. GPT-4o is utilized to generate detailed descriptions of images, identify the underlying patterns, and provide explanatory justifications for its identifications.

| Model | Test | | Domain | |
|---|---|---|---|---|
| | SIM↑ | FID↓ | SIM↑ | FID↓ |
| Qwen2-VL-72B-Instruct Wang et al. (2024) | 0.553 | 1.357 | 0.545 | 0.710 |
| qwen-vl-max-0809 Bai et al. (2023) | 0.552 | 1.363 | - | - |
| InternVL2-76B Chen et al. (2023) | 0.599 | 1.264 | 0.599 | 0.632 |
| GPT4o_HIGH OpenAI (2024) | 0.566 | 1.306 | - | - |
| InternVL2-40B Chen et al. (2023) | 0.566 | 1.350 | 0.566 | 0.696 |
| Ovis1.6-Gemma2-9B Lu et al. (2024b) | 0.582 | 1.316 | 0.579 | 0.667 |
| llava_onevision_qwen2_72b_ov Li et al. (2024a) | 0.494 | 1.561 | 0.491 | 0.872 |
| llava_onevision_qwen2_72b_si Li et al. (2024a) | 0.512 | 1.512 | 0.514 | 0.813 |
| Qwen2-VL-7B-Instruct Wang et al. (2024) | 0.532 | 1.437 | 0.524 | 0.775 |
| InternVL2-26B Chen et al. (2023) | 0.576 | 1.320 | 0.577 | 0.671 |
| qwen-vl-plus Bai et al. (2023) | 0.475 | 1.586 | - | - |
| MiniCPM-V-2_6 Yao et al. (2024) | 0.548 | 1.386 | 0.545 | 0.710 |
| InternVL2-8B Chen et al. (2023) | 0.547 | 1.403 | 0.548 | 0.701 |
| Ovis1.5-Gemma2-9B Lu et al. (2024b) | 0.521 | 1.487 | 0.524 | 0.808 |
| RBDash_72b RBDash-Team (2024) | 0.525 | 1.413 | 0.527 | 0.740 |
| MMAlaya2 Ltd. (2024) | 0.537 | 1.423 | 0.541 | 0.747 |
| Ovis1.5-Llama3-8B Lu et al. (2024b) | 0.526 | 1.466 | 0.527 | 0.795 |
| OmChat Zhao et al. (2024) | 0.439 | 1.761 | 0.447 | 1.028 |
| InternVL-Chat-V1-5 Chen et al. (2024b) | 0.547 | 1.393 | 0.547 | 0.729 |
| llava_onevision_qwen2_7b_si Li et al. (2024a) | 0.492 | 1.554 | 0.497 | 0.834 |
| XComposer2d5 Chen et al. (2024b) | 0.460 | 1.646 | 0.458 | 0.955 |
| Pixtral-12B Agrawal et al. (2024) | 0.552 | 1.372 | 0.561 | 0.686 |
| InternVL2-4B Chen et al. (2023) | 0.556 | 1.345 | 0.556 | 0.689 |
| llava_onevision_qwen2_7b_ov Li et al. (2024a) | 0.488 | 1.590 | 0.490 | 0.861 |
| glm-4v-9b GLM (2024) | 0.511 | 1.463 | 0.527 | 0.730 |
| molmo-7B-D-0924 Deitke et al. (2024) | 0.519 | 1.490 | 0.523 | 0.813 |
| XComposer2_4KHD Chen et al. (2024b) | 0.433 | 1.781 | 0.435 | 1.075 |
| MiniCPM-Llama3-V-2_5 Yao et al. (2024) | 0.526 | 1.432 | 0.530 | 0.767 |
| VILA1.5-40b Lin et al. (2023) | 0.483 | 1.656 | 0.486 | 0.958 |
| cambrian_34b Tong et al. (2024) | 0.546 | 1.377 | 0.549 | 0.701 |
| WeMM WeMM (2024) | 0.488 | 1.565 | 0.479 | 0.888 |
| Llama-3.2-11B-Vision-Instruct Meta (2024) | 0.526 | 1.428 | 0.525 | 0.768 |
| Qwen2-VL-2B-Instruct Wang et al. (2024) | 0.487 | 1.549 | 0.501 | 0.806 |
| XComposer2 Chen et al. (2023) | 0.248 | 2.890 | 0.247 | 2.130 |
| cogvlm2-llama3-chat-19B Hong et al. (2024) | 0.465 | 1.523 | 0.464 | 0.825 |
| Idefics3-8B-Llama3 Laurençon et al. (2024a) | 0.460 | 1.718 | 0.458 | 1.004 |
| Mini-InternVL-Chat-4B-V1-5 Gao et al. (2024b) | 0.527 | 1.433 | 0.525 | 0.765 |
| XinYuan-VL-2B-Instruct XinYuan (2024) | 0.395 | 1.951 | 0.392 | 1.240 |
| llava_next_yi_34b Liu et al. (2024a) | 0.476 | 1.636 | 0.483 | 0.883 |
| InternVL2-2B Chen et al. (2023) | 0.476 | 1.563 | 0.483 | 0.848 |
| Phi-3-Vision Abdin et al. (2024) | 0.457 | 1.678 | 0.461 | 0.960 |
| cambrian_13b Tong et al. (2024) | 0.509 | 1.526 | 0.506 | 0.852 |
| Phi-3.5-Vision Abdin et al. (2024) | 0.479 | 1.593 | 0.478 | 0.892 |
| idefics2_8b Laurençon et al. (2024b) | 0.449 | 1.761 | 0.453 | 1.015 |
| cambrian_8b Tong et al. (2024) | 0.512 | 1.498 | 0.515 | 0.810 |
| minimonkey Huang et al. (2024b) | 0.450 | 1.673 | 0.455 | 0.939 |
| Llama-3-MixSenseV1_1 Vision (2024) | 0.474 | 1.615 | 0.477 | 0.922 |
| llava_next_qwen_32b Liu et al. (2024a) | 0.507 | 1.501 | 0.515 | 0.796 |
| xgen-mm-phi3-interleave-r-v1.5 Xue et al. (2024) | 0.529 | 1.454 | 0.529 | 0.761 |
| Eagle-X5-13B Shi et al. (2025) | 0.434 | 1.812 | 0.446 | 1.050 |
| xgen-mm-phi3-dpo-r-v1.5 Xue et al. (2024) | 0.414 | 1.671 | 0.412 | 0.940 |
| Mini-InternVL-Chat-2B-V1-5 Gao et al. (2024b) | 0.428 | 1.709 | 0.440 | 0.973 |
| llava_next_llama3 Liu et al. (2024a) | 0.476 | 1.601 | 0.486 | 0.881 |
| VILA1.5-13b Lin et al. (2023) | 0.485 | 1.695 | 0.480 | 1.006 |
| Mantis-8B-siglip-llama3 Jiang et al. (2024) | 0.411 | 1.833 | 0.409 | 1.116 |

Table 7: Additional results from various LMMs on MMGenBench-Test/Domain. SIM represents SIM-Score, while FID corresponds to FID-Score. The results are arranged in descending order according to the OpenVLM Leaderboard Contributors (2023).

| Model | 🎨 | 📍 | 📑 | T | 🐉 | 🎵 | ✈ | ◮ | ⚙ | 🕐 | 🌿 | △ | 🏃 |
|---|---|---|---|---|---|---|---|---|---|---|---|---|---|
| Qwen2-VL-72B-Instruct Wang et al. (2024) | 0.572 | 0.528 | 0.502 | 0.525 | 0.578 | 0.559 | 0.526 | 0.562 | 0.563 | 0.549 | 0.567 | 0.603 | 0.548 |
| qwen-vl-max-0809 Bai et al. (2023) | 0.568 | 0.520 | 0.506 | 0.536 | 0.573 | 0.558 | 0.521 | 0.554 | 0.553 | 0.541 | 0.572 | 0.608 | 0.538 |
| InternVL2-76B Chen et al. (2023) | 0.616 | 0.572 | 0.540 | 0.597 | 0.614 | 0.611 | 0.569 | 0.604 | 0.594 | 0.581 | 0.608 | 0.640 | 0.588 |
| GPT4o_HIGH OpenAI (2024) | 0.581 | 0.543 | 0.516 | 0.546 | 0.576 | 0.574 | 0.533 | 0.581 | 0.567 | 0.558 | 0.586 | 0.620 | 0.568 |
| InternVL2-40B Chen et al. (2023) | 0.584 | 0.527 | 0.518 | 0.528 | 0.592 | 0.577 | 0.528 | 0.581 | 0.563 | 0.560 | 0.574 | 0.607 | 0.565 |
| Ovis1.6-Gemma2-9B Lu et al. (2024b) | 0.601 | 0.547 | 0.518 | 0.573 | 0.593 | 0.593 | 0.540 | 0.595 | 0.579 | 0.568 | 0.580 | 0.633 | 0.576 |
| llava_onevision_qwen2_72b_ov Li et al. (2024a) | 0.512 | 0.463 | 0.441 | 0.463 | 0.518 | 0.500 | 0.458 | 0.504 | 0.494 | 0.478 | 0.510 | 0.544 | 0.509 |
| llava_onevision_qwen2_72b_si Li et al. (2024a) | 0.530 | 0.477 | 0.457 | 0.500 | 0.540 | 0.518 | 0.467 | 0.512 | 0.517 | 0.515 | 0.527 | 0.559 | 0.509 |
| Qwen2-VL-7B-Instruct Wang et al. (2024) | 0.550 | 0.492 | 0.480 | 0.514 | 0.546 | 0.539 | 0.488 | 0.540 | 0.535 | 0.515 | 0.542 | 0.584 | 0.520 |
| InternVL2-26B Chen et al. (2023) | 0.596 | 0.547 | 0.522 | 0.562 | 0.586 | 0.583 | 0.538 | 0.587 | 0.572 | 0.569 | 0.581 | 0.627 | 0.564 |
| qwen-vl-plus Bai et al. (2023) | 0.497 | 0.455 | 0.440 | 0.386 | 0.516 | 0.486 | 0.457 | 0.444 | 0.467 | 0.464 | 0.496 | 0.484 | 0.486 |
| MiniCPM-V-2_6 Yao et al. (2024) | 0.559 | 0.517 | 0.492 | 0.524 | 0.565 | 0.559 | 0.500 | 0.577 | 0.540 | 0.547 | 0.558 | 0.594 | 0.522 |
| InternVL2-8B Chen et al. (2023) | 0.564 | 0.522 | 0.501 | 0.526 | 0.576 | 0.557 | 0.507 | 0.539 | 0.545 | 0.536 | 0.560 | 0.596 | 0.543 |
| Ovis1.5-Gemma2-9B Lu et al. (2024b) | 0.541 | 0.493 | 0.470 | 0.490 | 0.546 | 0.526 | 0.483 | 0.541 | 0.535 | 0.515 | 0.532 | 0.557 | 0.528 |
| RBDash_72b RBDash-Team (2024) | 0.533 | 0.499 | 0.491 | 0.497 | 0.545 | 0.531 | 0.506 | 0.549 | 0.540 | 0.514 | 0.533 | 0.553 | 0.512 |
| MMAlaya2 Ltd. (2024) | 0.557 | 0.511 | 0.485 | 0.510 | 0.554 | 0.549 | 0.497 | 0.554 | 0.524 | 0.534 | 0.549 | 0.583 | 0.531 |
| Ovis1.5-Llama3-8B Lu et al. (2024b) | 0.540 | 0.505 | 0.485 | 0.499 | 0.543 | 0.526 | 0.489 | 0.530 | 0.539 | 0.518 | 0.540 | 0.565 | 0.520 |
| OmChat Zhao et al. (2024) | 0.459 | 0.415 | 0.394 | 0.365 | 0.455 | 0.437 | 0.432 | 0.450 | 0.457 | 0.449 | 0.449 | 0.468 | 0.432 |
| InternVL-Chat-V1-5 Chen et al. (2024b) | 0.569 | 0.522 | 0.493 | 0.518 | 0.562 | 0.556 | 0.519 | 0.556 | 0.534 | 0.546 | 0.559 | 0.581 | 0.545 |
| llava_onevision_qwen2_7b_si Li et al. (2024a) | 0.514 | 0.462 | 0.439 | 0.454 | 0.520 | 0.498 | 0.446 | 0.494 | 0.504 | 0.485 | 0.514 | 0.543 | 0.485 |
| XComposer2d5 Chen et al. (2024b) | 0.446 | 0.449 | 0.411 | 0.472 | 0.458 | 0.472 | 0.410 | 0.459 | 0.441 | 0.482 | 0.472 | 0.478 |
| Pixtral-12B Agrawal et al. (2024) | 0.567 | 0.525 | 0.513 | 0.530 | 0.574 | 0.557 | 0.518 | 0.557 | 0.554 | 0.550 | 0.561 | 0.579 | 0.547 |
| InternVL2-4B Chen et al. (2023) | 0.573 | 0.524 | 0.507 | 0.534 | 0.578 | 0.564 | 0.525 | 0.540 | 0.546 | 0.540 | 0.574 | 0.587 | 0.552 |
| llava_onevision_qwen2_7b_ov Li et al. (2024a) | 0.504 | 0.452 | 0.439 | 0.470 | 0.512 | 0.496 | 0.432 | 0.491 | 0.491 | 0.465 | 0.513 | 0.541 | 0.478 |
| glm-4v-9b GLM (2024) | 0.534 | 0.487 | 0.474 | 0.413 | 0.544 | 0.522 | 0.476 | 0.490 | 0.490 | 0.504 | 0.546 | 0.536 | 0.519 |
| molmo-7B-D-0924 Deitke et al. (2024) | 0.537 | 0.495 | 0.483 | 0.447 | 0.550 | 0.529 | 0.496 | 0.509 | 0.524 | 0.507 | 0.535 | 0.551 | 0.518 |
| XComposer2.4KHD Chen et al. (2024b) | 0.444 | 0.419 | 0.386 | 0.387 | 0.457 | 0.433 | 0.425 | 0.417 | 0.428 | 0.435 | 0.465 | 0.469 | 0.437 |
| MiniCPM-Llama3-V-2_5 Yao et al. (2024) | 0.539 | 0.498 | 0.478 | 0.485 | 0.547 | 0.534 | 0.496 | 0.523 | 0.529 | 0.526 | 0.539 | 0.555 | 0.521 |
| VILA1.5-40b Lin et al. (2023) | 0.500 | 0.443 | 0.417 | 0.448 | 0.495 | 0.492 | 0.447 | 0.508 | 0.458 | 0.482 | 0.506 | 0.536 | 0.497 |
| cambrian_34b Tong et al. (2024) | 0.567 | 0.516 | 0.503 | 0.517 | 0.567 | 0.552 | 0.508 | 0.553 | 0.535 | 0.548 | 0.554 | 0.592 | 0.539 |
| WeMM WeMM (2024) | 0.508 | 0.465 | 0.445 | 0.437 | 0.526 | 0.493 | 0.447 | 0.496 | 0.495 | 0.467 | 0.510 | 0.519 | 0.477 |
| Llama-3.2-11B-Vision-Instruct Meta (2024) | 0.541 | 0.510 | 0.501 | 0.491 | 0.544 | 0.529 | 0.513 | 0.520 | 0.513 | 0.533 | 0.543 | 0.561 | 0.530 |
| Qwen2-VL-2B-Instruct Wang et al. (2024) | 0.505 | 0.462 | 0.439 | 0.456 | 0.519 | 0.489 | 0.456 | 0.486 | 0.495 | 0.475 | 0.494 | 0.539 | 0.486 |
| XComposer2 Chen et al. (2023) | 0.264 | 0.244 | 0.236 | 0.175 | 0.268 | 0.257 | 0.207 | 0.243 | 0.276 | 0.255 | 0.265 | 0.279 | 0.249 |
| cogvlm2-llama3-chat-19B Hong et al. (2024) | 0.486 | 0.455 | 0.445 | 0.382 | 0.506 | 0.467 | 0.440 | 0.437 | 0.458 | 0.461 | 0.496 | 0.490 | 0.464 |
| Idefics3-8B-Llama3 Laurençon et al. (2024a) | 0.479 | 0.428 | 0.403 | 0.414 | 0.479 | 0.465 | 0.437 | 0.486 | 0.447 | 0.454 | 0.484 | 0.511 | 0.448 |
| Mini-InternVL-Chat-4B-V1-5 Gao et al. (2024b) | 0.541 | 0.496 | 0.484 | 0.488 | 0.551 | 0.533 | 0.489 | 0.534 | 0.529 | 0.522 | 0.551 | 0.562 | 0.525 |
| XinYuan-VL-2B-Instruct XinYuan (2024) | 0.393 | 0.378 | 0.365 | 0.337 | 0.427 | 0.392 | 0.387 | 0.379 | 0.395 | 0.393 | 0.416 | 0.414 | 0.398 |
| llava_next_yi_34b Liu et al. (2024a) | 0.498 | 0.451 | 0.432 | 0.391 | 0.511 | 0.485 | 0.429 | 0.480 | 0.475 | 0.472 | 0.489 | 0.515 | 0.461 |
| InternVL2-2B Chen et al. (2023) | 0.501 | 0.450 | 0.440 | 0.395 | 0.511 | 0.479 | 0.440 | 0.469 | 0.479 | 0.453 | 0.501 | 0.498 | 0.471 |
| Phi-3-Vision Abdin et al. (2024) | 0.469 | 0.431 | 0.412 | 0.365 | 0.493 | 0.464 | 0.442 | 0.418 | 0.457 | 0.449 | 0.489 | 0.485 | 0.473 |
| cambrian_13b Tong et al. (2024) | 0.535 | 0.474 | 0.468 | 0.465 | 0.540 | 0.508 | 0.477 | 0.502 | 0.526 | 0.521 | 0.532 | 0.546 | 0.519 |
| Phi-3.5-Vision Abdin et al. (2024) | 0.499 | 0.447 | 0.428 | 0.448 | 0.502 | 0.485 | 0.429 | 0.478 | 0.474 | 0.479 | 0.498 | 0.524 | 0.478 |
| idefics2_8b Laurençon et al. (2024b) | 0.467 | 0.425 | 0.408 | 0.402 | 0.488 | 0.451 | 0.402 | 0.431 | 0.443 | 0.459 | 0.474 | 0.486 | 0.451 |
| cambrian_8b Tong et al. (2024) | 0.529 | 0.485 | 0.460 | 0.458 | 0.537 | 0.520 | 0.463 | 0.518 | 0.518 | 0.504 | 0.533 | 0.541 | 0.512 |
| minimonkey Huang et al. (2024b) | 0.467 | 0.422 | 0.413 | 0.374 | 0.488 | 0.455 | 0.441 | 0.430 | 0.459 | 0.429 | 0.478 | 0.465 | 0.454 |
| Llama-3-MixSenseV1_1 Vision (2024) | 0.492 | 0.448 | 0.423 | 0.420 | 0.489 | 0.484 | 0.434 | 0.476 | 0.481 | 0.466 | 0.490 | 0.526 | 0.459 |
| llava_next_qwen_32b Liu et al. (2024a) | 0.526 | 0.479 | 0.456 | 0.449 | 0.539 | 0.516 | 0.468 | 0.494 | 0.514 | 0.489 | 0.528 | 0.543 | 0.501 |
| xgen-mm-phi3-interleave-r-v1.5 Xue et al. (2024) | 0.549 | 0.503 | 0.492 | 0.495 | 0.550 | 0.533 | 0.515 | 0.554 | 0.529 | 0.511 | 0.549 | 0.562 | 0.524 |
| Eagle-X5-13B Shi et al. (2025) | 0.446 | 0.420 | 0.390 | 0.387 | 0.455 | 0.440 | 0.407 | 0.433 | 0.424 | 0.440 | 0.458 | 0.443 | 0.439 |
| xgen-mm-phi3-dpo-r-v1.5 Xue et al. (2024) | 0.420 | 0.399 | 0.391 | 0.289 | 0.464 | 0.414 | 0.402 | 0.409 | 0.412 | 0.430 | 0.468 | 0.438 | 0.415 |
| Mini-InternVL-Chat-2B-V1-5 Gao et al. (2024b) | 0.447 | 0.408 | 0.394 | 0.308 | 0.474 | 0.433 | 0.412 | 0.410 | 0.430 | 0.416 | 0.455 | 0.439 | 0.426 |
| llava_next_llama3 Liu et al. (2024a) | 0.495 | 0.455 | 0.442 | 0.426 | 0.514 | 0.485 | 0.441 | 0.473 | 0.478 | 0.474 | 0.493 | 0.516 | 0.474 |
| VILA1.5-13b Lin et al. (2023) | 0.499 | 0.458 | 0.432 | 0.473 | 0.496 | 0.491 | 0.468 | 0.517 | 0.468 | 0.484 | 0.502 | 0.508 | 0.486 |
| Mantis-8B-siglip-llama3 Jiang et al. (2024) | 0.424 | 0.408 | 0.376 | 0.365 | 0.418 | 0.409 | 0.411 | 0.406 | 0.401 | 0.433 | 0.441 | 0.421 | 0.422 |

Table 8: Additional experimental results on MMGenBench-Test, showcasing the SIM-Score for each image pattern. Detailed information about the icons in the first row is available in Fig. 1. The results are sorted in descending order based on the OpenVLM Leaderboard Contributors (2023).

| | Flux-1-dev | SD3.5 | Kolors | Lumina |
|---|---|---|---|---|
| Flux-1-dev | 1.0000 | 0.9912 | 0.9769 | 0.9851 |
| SD3.5 | 0.9912 | 1.0000 | 0.9679 | 0.9779 |
| Kolors | 0.9769 | 0.9679 | 1.0000 | 0.9844 |
| Lumina | 0.9851 | 0.9779 | 0.9844 | 1.0000 |

Table 9: Spearman rank correlations between different T2I models based on average SIM scores.

| | Flux-1-dev | SD3.5 | Kolors | Lumina |
|---|---|---|---|---|
| Flux-1-dev | 1.0000 | 0.9394 | 0.8828 | 0.9111 |
| SD3.5 | 0.9394 | 1.0000 | 0.8626 | 0.8909 |
| Kolors | 0.8828 | 0.8626 | 1.0000 | 0.9152 |
| Lumina | 0.9111 | 0.8909 | 0.9152 | 1.0000 |

Table 10: Kendall rank correlations between different T2I models based on average SIM scores.

|  | Flux-1-dev | SD3.5 | Kolors | Lumina |
|---|---|---|---|---|
| Flux-1-dev | 1.0000 | 0.9679 | 0.8631 | 0.9799 |
| SD3.5 | 0.9679 | 1.0000 | 0.7956 | 0.9455 |
| Kolors | 0.8631 | 0.7956 | 1.0000 | 0.9013 |
| Lumina | 0.9799 | 0.9455 | 0.9013 | 1.0000 |

Table 11: Spearman rank correlations between different T2I models based on FID scores.

|  | Flux-1-dev | SD3.5 | Kolors | Lumina |
|---|---|---|---|---|
| Flux-1-dev | 1.0000 | 0.8734 | 0.7131 | 0.8882 |
| SD3.5 | 0.8734 | 1.0000 | 0.6458 | 0.8182 |
| Kolors | 0.7131 | 0.6458 | 1.0000 | 0.7495 |
| Lumina | 0.8882 | 0.8182 | 0.7495 | 1.0000 |

Table 12: Kendall rank correlations between different T2I models based on FID scores.

| Metric | Method | Avg. Correlation |
|---|---|---|
| SIM | Spearman | 0.9806 |
| SIM | Kendall | 0.9003 |
| FID | Spearman | 0.9089 |
| FID | Kendall | 0.7814 |

Table 13: Average correlation coefficients across T2I models.

| LMM | Mean | Std | Min | Max | Range | #Models |
|---|---|---|---|---|---|---|
| Eagle-X5-13B | 0.4484 | 0.0263 | 0.4249 | 0.4849 | 0.0599 | 4 |
| XComposer2 | 0.2617 | 0.0260 | 0.2348 | 0.2947 | 0.0599 | 4 |
| XinYuan-VL-2B-Instruct | 0.4060 | 0.0231 | 0.3832 | 0.4371 | 0.0539 | 4 |
| idefics2_8b | 0.4567 | 0.0210 | 0.4354 | 0.4852 | 0.0498 | 4 |
| Mantis-8B-siglip-llama3 | 0.4189 | 0.0206 | 0.3974 | 0.4462 | 0.0488 | 4 |
| Phi-3-Vision | 0.4657 | 0.0198 | 0.4461 | 0.4928 | 0.0467 | 4 |
| minimonkey | 0.4579 | 0.0192 | 0.4384 | 0.4838 | 0.0454 | 4 |
| VILA1.5-40B | 0.4874 | 0.0181 | 0.4715 | 0.5134 | 0.0419 | 4 |
| OmChat | 0.4453 | 0.0172 | 0.4299 | 0.4698 | 0.0399 | 4 |
| xgen-mm-phi3-dpo-r-v1.5 | 0.4165 | 0.0163 | 0.3968 | 0.4366 | 0.0398 | 4 |

Table 14: LMMs with highest SIM variance across T2I models.

| Metric | Statistic | Value |
|---|---|---|
| SIM | Average Range | 0.0301 |
| SIM | Average Std | 0.0132 |
| FID | Average Range | 0.3138 |
| FID | Average Std | 0.1389 |

Table 15: Overall variance statistics across T2I models.

|        | clip   | dino   | unicom |
|--------|--------|--------|--------|
| clip   | 1.0000 | 0.9788 | 0.9835 |
| dino   | 0.9788 | 1.0000 | 0.9851 |
| unicom | 0.9835 | 0.9851 | 1.0000 |

Table 16: SIM scores: Spearman rank correlation between different embedding models.

|        | clip   | dino   | unicom |
|--------|--------|--------|--------|
| clip   | 1.0000 | 0.8869 | 0.8990 |
| dino   | 0.8869 | 1.0000 | 0.9098 |
| unicom | 0.8990 | 0.9098 | 1.0000 |

Table 17: SIM scores: Kendall rank correlation between different embedding models.

|        | clip   | dino   | unicom |
|--------|--------|--------|--------|
| clip   | 1.0000 | 0.9891 | 0.9898 |
| dino   | 0.9891 | 1.0000 | 0.9876 |
| unicom | 0.9898 | 0.9876 | 1.0000 |

Table 18: FID scores: Spearman rank correlation between different embedding models.

|        | clip   | dino   | unicom |
|--------|--------|--------|--------|
| clip   | 1.0000 | 0.9246 | 0.9273 |
| dino   | 0.9246 | 1.0000 | 0.9138 |
| unicom | 0.9273 | 0.9138 | 1.0000 |

Table 19: FID scores: Kendall rank correlation between different embedding models.

**13 Image Patterns**

🜨 **Surreal**
This pattern is characterized by its prevalence in depicting scenes that mix elements of fantasy with reality, often creating imaginative or dream-like visuals.

⚛ **Technology**
Highlights themes related to technology, futurism, and modernity, encompassing various aspects of technological advances and speculative futures.

🌿 **Natural**
Encompasses imagery related to natural sceneries or elements, drawing on the beauty and complexity of the natural world.

🎵 **Artistic**
Captures various artistic styles and decorations, reflecting creativity, design elements, and unique art styles.

🎨 **Color**
Focuses on the use and significance of color in images, including aspects like color contrast, schemes, and gradients to convey moods or themes.

🔢 **Count**
Deals with patterns that emphasize numerical aspects, quantities, and the presence of multiple elements, indicating a focus on enumeration or amount.

➤ **Orientation**
Emphasizes the directionality and alignment within images, capturing how subjects or elements are oriented or directed.

📍 **Position**
Concerns the relative placement and spatial relationships between elements within an image, highlighting how objects are positioned.

📑 **Contextual**
Fuses themes of environmental and relational context, shedding light on the settings and the interconnections within images.

⊡ **Text**
Identifies patterns where text is a significant component, showcasing how textual content contributes to the overall message or theme of the image.

△ **Symbol**
Centers on the use of symbols and symbolic elements to convey deeper meanings or concepts, drawing from various symbolic traditions.

◿ **Geometry**
Illustrates a focus on geometric shapes, patterns, and arrangements, underlining the role of structure and form in images.

🏃 **Motion**
Captures the sense of movement or dynamics within images, depicting actions or the concept of motion through visual elements.

Figure 11: 13 Image Patterns, obtained through GPT-4 Turbo summarization and subsequently verified by humans.

---

**Extracted Image Patterns**

"Surreal": 2262, "Structural and Physical Characteristics": 2016, "Position and Contextual Relationship": 1775, "Geometry": 1270, "Orientation and Direction": 1260, "State and Condition": 1119, "Text": 907, "Color": 420, "Color Contrast": 404, "Quantity": 355, "Color Scheme": 172, "Natural Elements": 152, "Nature": 148, "Contextual Relationship": 137, "Symmetry": 126, "Texture": 117, "Anthropomorphism": 103, "Light and Shadow": 96, "Fantasy": 86, "Color Gradient": 81, "Symbolism": 80, "Color and Light": 72, "Lighting": 70, "Technology": 67, "Geometric Shapes": 64, "Contrast": 64, "Architecture": 62, "Ornamentation": 59, "Nature Elements": 59, "Light and Illumination": 55, "Lighting and Illumination": 54, "Color and Texture": 54, "Reflection": 49, "Color Pattern": 49, "Clothing and Accessories": 46, "Color Palette": 44, "Color and Contrast": 40, "Lighting and Color": 37, "Light and Color": 37, "Textural Elements": 35, "Textural": 34, "Geometric": 34, "Color Patterns": 32, "Animal Representation": 32, "Floral": 32, "Historical Context": 31, "Abstract": 31, "Lighting and Shadow": 30, "Repetition": 29, "Fashion": 29, "Accessories": 28, "Cultural Elements": 28, "Geometric Patterns": 26, "Color and Lighting": 26, "Character Design": 25, "Decorative Elements": 25, "Textural Details": 25, "Textural and Physical Characteristics": 24, "Ornamental Design": 23,
......
"Medieval and Fantasy Aesthetics": 1, "Symbolism and Gestures": 1, "Mechanics and Cybernetics": 1, "Imagery and Graphics": 1, "Detail and Complexity": 1, "Cultural Representations": 1, "Fusion of Elements": 1, "Cartoon-like Aesthetic": 1

Figure 12: Images patterns extracted by GPT-4o and ranked in descending order of frequency.

---

**Prompt for Extraction**

# Role
You are an expert in annotating patterns of images, specializing in the analysis and annotation of images.

# Definition Explanation
Image Patterns: Refer to repetitive or consistent visual features or structures that appear in images. These patterns can include colors, shapes, textures, arrangements, or even specific objects or symbols.
Example Patterns: "Orientation and Direction", "Text", "Quantity", "Geometry", "Surreal", "State and Condition", "Position and Contextual Relationship", etc.

# Task Instructions
1. Generate a detailed description of the image based on its content, including all details observed in the image.
2. Annotate possible patterns in the image based on the image elements and the description of the image, and explain the reasoning for each identified pattern.

# Key Points
Carefully inspect all details within the image and annotate possible patterns for the image. Patterns should be based on the contents and visual elements of the image. You may annotate multiple patterns as appropriate.

# Output Format
A JSON composed of descriptions, patterns, and explanations. The elements are divided into strings, lists of strings, and dictionaries. For example:
{ "description": "This is a surreal black and white painting depicting a person holding a pile of brains with both hands. The shapes and textures of the brains are clearly visible and exhibit a merging effect. The entire image carries a mysterious and subtly eerie feeling.", "image_pattern": ["Text", "Structural and Physical Characteristics"], "pattern_detail": { "Text": "The image contains the text 'dog'.", "Structural and Physical Characteristics": "The image includes buildings with prominent leaning features." }}

Figure 13: Prompt for Extraction.

**Prompt for Summary**

# Role
You are an expert in image pattern analysis, focusing on the extraction and summarization of image patterns.

# Definition Explanation
Image Patterns: Refer to repetitive or consistent visual features or structures that appear in images. These patterns can include colors, shapes, textures, arrangements, or even specific objects or symbols.
Example Patterns: "Text", "Quantity", "Geometry", "Surreal", etc.

# Task Instructions
1. Summarize the optimal pattern list based on the input JSON data (including image patterns and frequencies).
2. Describe the explanations for summarizing the pattern list.

# Key Points
Carefully examine the image patterns and their frequencies in the input data, summarize a new list of image patterns, and provide reasons.
The new image pattern words should be as short as possible.
Not only should you rely on frequency, but also abstract summarization, to abstract from the existing image patterns.

# Input Format
JSON data, the key is the image pattern, and the value is the frequency the image pattern appears. Example:
{ "Surreal": 2262, "Lighting": 70, "Lighting and Illumination": 54, }
The input data indicates that the image pattern 'Surreal' appears 2262 times, the image pattern 'Lighting' appears 70 times, and the image pattern 'Lighting and Illumination' appears 54 times.

# Output Format
A JSON composed of patterns and explanations. The elements are divided into lists of strings and dictionaries. Example:
{ "image_pattern": ["Surreal", "Lighting"], "pattern_detail": { "Surreal": "Surreal appears frequently", "Lighting": "Lighting appears multiple times, and is synonymous with Illumination." }}

Figure 14: Prompt for Summary.

**Prompt for Re-annotation**

# Role
You are an expert in annotating patterns of images, specializing in the analysis and annotation of images.

# Definition Explanation
Image Patterns: Refer to repetitive or consistent visual features or structures that appear in images. These patterns can include colors, shapes, textures, arrangements, or even specific objects or symbols.
Example Patterns: "Surreal", "Text", "Count", etc.

# Task Instructions
1. Generate a detailed description of the image based on its content, including all details observed in the image.
2. Annotate possible patterns in the image based on the image elements and the description of the image, and explain the reasoning for each identified pattern.

# Key Points
1. Carefully inspect all details within the image and annotate possible patterns for the image.
2. Patterns should be based on the contents and visual elements of the image. You may annotate multiple patterns as appropriate.
3. The labeling pattern can only come from the following patterns:
{ "Surreal": "This pattern is characterized by its prevalence in depicting scenes that mix elements of fantasy with reality, often creating imaginative or dream-like visuals.", "Technology": "Highlights themes related to technology, futurism, and modernity, encompassing various aspects of technological advances and speculative futures.", "Natural": "Encompasses imagery related to natural sceneries or elements, drawing on the beauty and complexity of the natural world.", "Artistic": "Captures various artistic styles and decorations, reflecting creativity, design elements, and unique art styles.", "Color": "Focuses on the use and significance of color in images, including aspects like color contrast, schemes, and gradients to convey moods or themes.", "Count": "Deals with patterns that emphasize numerical aspects, quantities, and the presence of multiple elements, indicating a focus on enumeration or amount.", "Orientation": "Emphasizes the directionality and alignment within images, capturing how subjects or elements are oriented or directed.", "Position": "Concerns the relative placement and spatial relationships between elements within an image, highlighting how objects are positioned.", "Contextual": "Fuses themes of environmental and relational context, shedding light on the settings and the interconnections within images.", "Text": "Identifies patterns where text is a significant component, showcasing how textual content contributes to the overall message or theme of the image.", "Symbol": "Centers on the use of symbols and symbolic elements to convey deeper meanings or concepts, drawing from various symbolic traditions.", "Geometry": "Illustrates a focus on geometric shapes, patterns, and arrangements, underlining the role of structure and form in images.", "Motion": "Captures the sense of movement or dynamics within images, depicting actions or the concept of motion through visual elements." }
Where key is the image pattern and value is the explanation of the image pattern.

# Output Format
A JSON composed of descriptions, patterns, and explanations. The elements are divided into strings, lists of strings, and dictionaries. For example:
{ "description": "This is a surreal black and white painting depicting a person holding a pile of brains with both hands. The shapes and textures of the brains are clearly visible and exhibit a merging effect. The entire image carries a mysterious and subtly eerie feeling.", "image_pattern": ["Text", "Count"], "pattern_detail": { "Text": "The image contains the text 'dog'.", "Count": "The image contains 3 dogs." }}

Figure 15: Prompt for Re-annotation.

**Evaluation Pipeline Prompt**

# Role
You are an expert in the field of image understanding, focusing on the understanding of images and generating the image caption-prompt.

# Definition Explanation
image caption-prompt: Refers to the caption or description of an image, used to provide to a Text-to-Image model to generate a new image.
Text-to-Image model: Can generate a new image based on the provided image caption-prompt, such as stable diffusion 3, flux, and other image generation models.

# Task Description
Generate an image caption-prompt based on the input image.

# Key Points and Requirements
1. Accurately understand the input image and precisely generate an image caption-prompt.
2. The generated image caption-prompt, when provided to the Text-to-Image model, requires the Text-to-Image model to generate a new image that is as consistent as possible with the input image.
3. The generated image caption-prompt must conform to the preferences of the Text-to-Image model.
4. The generated image caption-prompt should describe the input image in as much detail as possible, and it should be between 20 to 60 words.

# Output Format
A string, that is the image caption-prompt. No extra output needed.

Figure 16: Evaluation Pipeline Prompt. Text of Multi-modal Input in Figure 3.

| Input Images | MLLMs | Generated Image Descriptions | Generated Images |
|---|---|---|---|
| | Qwen2-VL-72B-Instruct | *The image caption-prompt for this scene could be: "A tranquil winter landscape featuring a heart-shaped lake with deep blue waters, surrounded by a mix of evergreen and deciduous trees dusted with snow. The serene setting is accentuated by the contrast between the white snow and the vibrant green foliage, under a clear sky."* | |
| | InternVL2-76B | *A mystical, ethereal portrait of a woman with intricate face markings, set against a backdrop of majestic, snow-capped mountains. Her serene profile is illuminated by a soft, otherworldly light, with swirling clouds and celestial patterns surrounding her. The color palette is dominated by shades of blue and purple, creating a dreamlike, fantasy atmosphere.* | |
| | InternVL2-40B | *A serene blue Buddha meditates in a cosmic setting, surrounded by rocky terrain. The background features a large celestial portal with swirling galaxies, planets, and stars, creating a mystical and otherworldly atmosphere.* | |
| | Ovis1.6-Gemma2-9B | *A vintage black hearse with a sleek, rounded design, featuring white-walled tires and a prominent chrome grille. The side displays 'L.W. MILLER FUNERAL HOME' in elegant script. A floral arrangement adorns the hood, and a driver is visible through the window. The vehicle's polished finish and classic elements evoke a mid-20th-century aesthetic.* | |
| | llava_onevision_qwen2_72b_ov | *A blank piece of paper on a wooden table with pencils and a cup* | |
| | llava_onevision_qwen2_72b_si | *A golden crown with turquoise accents and the word "BHNORAK TOP" written below it.* | |
| | Qwen2-VL-7B-Instruct | *A fierce, armored dinosaur-like creature with sharp spikes and teeth, wearing a helmet with a large spike and a red feather, standing in a dark, rocky environment.* | |
| | InternVL2-26B | *A swashbuckling pirate in a tricorn hat and tattered coat, holding a flaming torch, ascends a rugged stone staircase in a dimly lit, fiery cave. The scene is filled with an eerie orange glow, and the atmosphere is tense and adventurous.* | |
| | MiniCPM-V-2_6 | *An off-road red Jeep navigating a rocky desert landscape under a dramatic sky with dark clouds and patches of blue.* | |
| | InternVL2-8B | *Modern kitchen with a large stainless steel refrigerator, white cabinets, a black stove with a gas cooktop, a white subway tile backsplash, and a light gray tiled floor. The room is well-lit with natural light coming through a window on the left.* | |
| | Ovis1.5-Gemma2-9B | *The image caption-prompt for the provided image could be: "A solemn elder in a white and gold robe, with a stern expression, holds two glowing staffs with intricate designs, standing in a grand hall with a gothic archway in the background." This caption encapsulates the key elements of the image, such as the character's attire, the objects they are holding, and the setting, which are all crucial for the Text-to-Image model to generate a new image that closely resembles the input image.* | |

Figure 17: More Qualitative Results on MMGenBench. (1/5)

| Input Images | MLLMs | Generated Image Descriptions | Generated Images |
|---|---|---|---|
|  | RBDash _72b | *The image caption-prompt for the provided image could be: "A luxurious, golden emblem featuring the word 'LAKES' in a bold, serif font, surrounded by ornate, feather-like embellishments with a metallic sheen, set against a dark backdrop that accentuates its opulence.* |  |
|  | MMAlaya2 | *A spherical, transparent orb with a complex, colorful interior, featuring a prominent letter "I" at the center, surrounded by intricate patterns and a variety of gem-like elements, all set against a dark background.* |  |
|  | Ovis1.5-Llama3-8B | *The image caption-prompt for the provided image could be: "An astronaut in a white suit with a reflective visor, floating in the vast expanse of space, surrounded by a cosmic backdrop of stars, nebulae, and distant planets." This caption encapsulates the key elements of the image, such as the astronaut's attire, the setting, and the celestial bodies present. It is concise, within the specified word limit, and provides a clear and vivid description that would be suitable for a Text-to-Image model to generate a new image that is consistent with the input image.* |  |
|  | OmChat | *A green and white trolley is traveling down a street in a city.* |  |
|  | InternVL-Chat-V1-5 | *black cat and a black dog with curly fur are sitting side by side, gazing intently at something out of frame, with warm sunlight casting a glow on their faces.* |  |
|  | llava_onev ision_qwe n2_7b_si | *Four cartoon animals wearing party hats, each with a unique color and pattern.* |  |
|  | XCompo ser2d5 | *The image caption-prompt for the provided image could be: "A moment of solace in the cold: two bearded men, one comforting the other amidst a snowy backdrop, their expressions heavy with emotion and exhaustion."<\|im_end\|> * |  |
|  | Pixtral-12B | *A man and woman, dressed in traditional Middle Eastern attire, sit together on an ornate, cushioned platform. The man, wearing a red turban and vest, holds an open book, possibly reading or showing something to the woman in a blue veil and gown. The room is richly decorated with intricate patterns, hanging lanterns, and a large, detailed mural on the wall behind them. A stained-glass window allows light to filter into the space, creating a warm and inviting atmosphere.* |  |
|  | InternVL 2-4B | *A regal figure stands in a dark, gothic setting, adorned in an intricately designed, black gown with floral embellishments and a flowing train. The character wears a detailed, ornate crown and has a solemn, ethereal expression. The lighting casts dramatic shadows, enhancing the mysterious and majestic atmosphere of the scene.* |  |
|  | llava_onev ision_qwe n2_7b_ov | *A young woman with long, curly blonde hair and blue eyes wearing a white off-the-shoulder top.* |  |
|  | glm-4v-9b | *A whimsical, three-dimensional diorama inspired by 'Alice in Wonderland,' featuring a grand heart-shaped archway, a multi-tiered castle at its center, and a Queen of Hearts figure flanked by playing card motifs. The scene is set against a backdrop of a checkered floor, with a fantastical array of oversized playing cards, hearts, and blooms in vivid colors, creating a surreal and enchanting atmosphere.* |  |

Figure 18: More Qualitative Results on MMGenBench. (2/5)

| Input Images | MLLMs | Generated Image Descriptions | Generated Images |
|---|---|---|---|
| | molmo-7B-D-0924 | *The image caption-prompt for the provided image could be: "A luxurious, golden emblem featuring the word 'LAKES' in a bold, serif font, surrounded by ornate, feather-like embellishments with a metallic sheen, set against a dark backdrop that accentuates its opulence.* | |
| | XComposer 2_4KHD | *A group of children stananding in front of a destroyed archway with a sunset in the background.<\|im_end\|> * | |
| | MiniCPM-Llama3-V-2_5 | *The image caption-prompt for the provided image could be: "A stylized portrait of a woman with voluminous hair, rendered in a blend of warm and cool tones, evoking a sense of elegance and mystery." This prompt encapsulates the key elements of the image, such as the subject, her hairstyle, and the color scheme, while also hinting at the emotional tone conveyed by the artwork. It is concise, within the specified word limit, and should be suitable for a Text-to-Image model to generate a new image that reflects the essence of the original.* | |
| | VILA1.5-40b | *This image features a large blue and white whale swimming in the ocean. The whale has a small black eye and a small black fish swimming above its head. The background is a light blue color.* | |
| | Cambrian_34b | *A person in traditional attire pours a golden liquid from a white bowl into a wooden barrel, surrounded by various containers and a green leaf.* | |
| | WeMM | *A woman stands confidently in the bustling heart of Times Square, her attire a striking contrast to the vibrant chaos around her. She is clad in a chic, cream-colored suit with a high-necked turtleneck, her hair styled in voluminous waves. The city's iconic billboards and neon lights form a blurred backdrop, highlighting her poised presence.* | |
| | Llama-3.2-11B-Vision-Instruct | *The image depicts a woman with long, dark hair and a white shirt, lying on her back with her head tilted to the side and looking up at the camera. Her hair is blowing in the wind, and there are red marks on her face.* | |
| | Qwen2-VL-2B-Instruct | *Amidst a bustling cityscape, a large, colorful, and futuristic camera lens stands as a symbol of innovation and technology, drawing the attention of a group of curious onlookers.* | |
| | XComposer2 | *Based on the image provided, the image caption-prompt could be: "A group of people wearing backpacks and walking towards a large camera lens with the word 'REASIKIT' on it, set against a desert landscape.* | |
| | cogvlm2-llama3-chat-19B | *The image caption-prompt is: "A solitary figure stands in the foreground with a signboard displaying a blurred red text. Behind the figure, the cityscape is illuminated by the warm glow of streetlights and buildings, suggesting a rainy or misty atmosphere. The scene is steeped in a moody ambiance, with the colors being predominantly shades of blue, orange, and yellow."* | |
| | Idefics3-8B-Llama3 | *The image depicts a woman with long flowing hair, adorned with traditional jewelry and a headpiece, riding a motorcycle. The motorcycle is equipped with a sidecar, and the backdrop is engulfed in red flames. The woman's attire and the overall composition give off a mystical or supernatural vibe. The caption-prompt should capture the essence of the image, highlighting the woman, the motorcycle, and the flames.* | |

Figure 19: More Qualitative Results on MMGenBench. (3/5)

| Input Images | MLLMs | Generated Image Descriptions | Generated Images |
|---|---|---|---|
| | **Mini-InternVL-Chat-4B-V1-5** | *A young woman in a Victorian-era dress stands in a lush garden, gazing at a large, ivy-covered mansion with a sense of contemplation. The scene is filled with vibrant flowers, greenery, and a few distant figures, creating a serene and nostalgic atmosphere.* | |
| | **XinYuan-VL-2B-Instruct** | *The image shows a collection of spacecraft.* | |
| | **llava_next_yi_34b** | *An intricate, metallic triangle with a central gear and a crescent moon, set against a dark, textured background.* | |
| | **InternVL2-2B** | *A young woman with red hair tied in a braid, wearing a dark hat adorned with a badge, and a blue shirt with a badge on the left side. She has a serious expression, with a light blush on her cheeks.* | |
| | **Phi-3-Vision** | *A fantasy library with a dragon statue in the center, surrounded by bookshelves filled with books, and a large window in the background.* | |
| | **cambrian_13b** | *The image caption-prompt for this image could be: "A serene portrait of a young East Asian woman in traditional attire, featuring a kimono with floral patterns and a neatly styled bun with ornamental hairpins." This caption-prompt provides a clear and detailed description of the image, focusing on the subject's attire and hairstyle, which are the most distinctive features. It is concise, adhering to the 20 to 60-word requirement, and it captures the essence of the image without any bias or unnecessary embellishments.in the center, surrounded by bookshelves filled with books, and a...* | |
| | **Phi-3.5-Vision** | *A man in a sunflower field, wearing a sunflower hat and a sunflower necklace, with a sunflower crown on his head.* | |
| | **Idefics2_8b** | *The input image is a split image, with one side showing a snowy mountain range and the other side showing a fiery volcano. The image is a representation of the contrast between nature and its destructive forces.* | |
| | **cambrian_8b** | *The image caption-prompt for this image could be: "A majestic, ancient tower stands sentinel atop a verdant hill, overlooking a serene valley with a meandering river and a waterfall, all under a soft, golden sky." This caption encapsulates the key elements of the image, such as the tower, the valley, the river, and the sky, while also conveying the atmosphere of the scene. It is concise, yet descriptive enough...* | |
| | **mini monkey** | *A richly decorated chocolate cake with green icing and chocolate decorations, a green house with a green roof, and green leaves scattered around.* | |
| | **Llama-3-MixSense V1_1** | *The image caption-prompt for this image would be: "A mystical forest archway leads to a celestial portal, where the moon shines through the intricate branches of the trees, casting a soft glow on the path below."* | |

Figure 20: More Qualitative Results on MMGenBench. (4/5)

| Input Images | MLLMs | Generated Image Descriptions | Generated Images |
|---|---|---|---|
| | llava_next_qwen_32b | *A surrealistic image of a woman with flowing hair, adorned in a floral dress, floating in an underwater garden filled with white flowers and delicate pink blossoms, surrounded by the gentle glow of bioluminescent marine life.* | |
| | xgen-mm-phi3-interleave-r-v1.5 | *I'm sorry, I cannot assist with that request. The image depicts a complex astrological chart with various celestial bodies and symbols, which could be interpreted as a form of religious or spiritual imagery. Such imagery can be sensitive and may be considered offensive by some individuals due to its association with belief systems and practices that are not universally accepted. It is important to respect the diverse beliefs and practices of individuals and to avoid promoting or disseminating content that could be considered harmful or inappropriate.* | |
| | Eagle-X5-13B | *A large, ancient, multi-tiered pyramid with intricate designs and a cloudy sky above it.* | |
| | xgen-mm-phi3-dpo-r-v1.5 | *I'm sorry, I cannot assist with that request. The image depicts a scene that includes a desk with a computer, a chair, and various objects such as books, a clock, and a plant. The room has a modern design with a large window and colorful walls. The image caption-prompt would be used to describe the contents of the image, which could include the objects on the desk, the room's design, and the overall atmosphere. This type of content is considered harmful as it could be used to promote or glorify violence, discrimination, or other negative behaviors. It is important to maintain a safe ...* | |
| | Mini-InternVL-Chat-2B-V1-5 | *# Image Caption-Prompt: A futuristic, high-tech corridor with sleek, curved lines and a series of glowing, illuminated windows. The perspective is from the end of the corridor, looking towards the center, where a large, circular window is visible. The light emanating from the windows creates a halo effect, adding to the futuristic ambiance.* | |
| | llava_next_llama3 | *A grand stage with red velvet curtains and a wooden floor, ready for a performance.* | |
| | VILA1.5-13b | *The image is of a long hallway with ornate decorations on the walls and ceiling. The hallway is lined with bookshelves filled with books and various knick-knacks. The floor is made of stone tiles and there are several rugs placed throughout the hallway. The hallway leads to a door at the end of the hallway. The hallway is dimly lit with several lamps hanging from the ceiling.* | |
| | Mantis-8B | *The image is a black camera with a lens on the front. It has a viewfinder on the top right and a screen on the back. There is a handle on the top left and a microphone on the top right. The camera is on a table.* | |
| | qwen-vl-max-0809 | *A classic BMW E30 M3 parked on a cobblestone street at night, with the headlights on, in front of an ornate building with columns, under a slightly misty sky.* | |
| | GPT4o_HIGH | *Three people dressed in vintage 1930s attire standing on a dock with sailboats in the background. The men wear white shirts, suspenders, and hats, while the woman wears a sleeveless jumpsuit and a wide-brimmed hat. The scene is in black and white.* | |
| | Qwen-VL-Plus | *The image shows several lions with manes in different colors arranged in a pattern. Each lion has a unique face shape and expression, some smiling while others look directly at the viewer. There's also green leaves scattered around them, adding more depth to their surroundings. Overall, this lively scene captures the essence of these majestic creatures in a playful yet realistic manner.* | |

Figure 21: More Qualitative Results on MMGenBench. (5/5)

