# OpenReview forum: "MMGenBench: Fully Automatically Evaluating LMMs from the Text-to-Image Generation Perspective"
_ICLR.cc/2026/Conference — Submitted to ICLR 2026_

### Official Review · Reviewer_iYF5 · 2025-10-27

**Soundness:** 2
**Presentation:** 2
**Contribution:** 2
**Rating:** 4
**Confidence:** 3

**Summary:**

The paper introduces MMGenBench, a fully automated pipeline for evaluating LMMs’ image understanding and detailed description. Given an input image, an LMM generates a fine-grained caption; a text-to-image (T2I) model then synthesizes an auxiliary image from that text; finally, a single image encoder (Unicom) computes feature-level similarity (SIM, FID) between the original and synthesized images. Two datasets are provided: MMGenBench-Test (13 image patterns, 1,284 images) and MMGenBench-Domain (~10k images). Results are reported for 50+ LMMs; the best SIM is < 0.6, and a human-alignment study reports 88.27% agreement on 1,850 pairs.

**Strengths:**

1.Clear, modular pipeline with explicit metric definitions (SIM/FID).

2.Broad coverage (50+ LMMs) with series-wise and pattern-wise analyses.

3.Practical cross-domain evaluation requiring only images.

**Weaknesses:**

1.Metric robustness not established.  Features come from a single encoder (Unicom), and main results default to one T2I model (FLUX.1-dev); no ablations on encoder/T2I/seed variation or rank correlations are reported.

2.Length/style bias. Shorter descriptions tend to receive lower scores.

3.Stochasticity not controlled. Evidence: Randomness is acknowledged (Eq. 2), but no multi-run statistics or confidence intervals are provided.

4.Human alignment under-analyzed. Only a single overall figure (88.27%) is given;

**Questions:**

1.What is the per-model seed variance of SIM?

---

> ### Author Response · Authors · 2025-11-28
> **To Reviewer iYF5**
>
> ## To Q1 & W3: Seed Variance
>
> We appreciate the reviewer’s concern regarding stochasticity.
> To quantify variance, we conducted a **five seed evaluation** (0, 23, 42, 78, 1234) on the MMGenBench-Test and report the mean and std of SIM and FID for 23 main models.
>
> Across all 23 LMMs, the stochastic variation is **minimal**:
>
> * **SIM std:** 0.0002–0.0038
> * **FID std:** 0.0022–0.0141
> * **Model ranking is invariant across seeds** (SIM Spearman = 0.9975, FID Spearman = 0.9970)
>
> These results show that multi run fluctuations are negligible, and the scores in Table 1 (in the paper) closely match the multi-seed means (difference ≤ 0.004). The evaluation pipeline is therefore **stable and robust** to T2I sampling randomness.
>
> ---
>
> ## To W1: External Model Dependency
>
> We thank the reviewer for highlighting this important point. To quantify the influence of auxiliary models, we conducted systematic sensitivity analyses across **four T2I models** (FLUX-1-dev, SD3.5, Kolors, Lumina) and **three widely used encoders** (CLIP-L/336, DINOv2-L, Unicom-L/336). (Please see **#aasA/W1, #aasA/W2** for full results.)
>
> **T2I robustness.**
> LMM rankings remain highly consistent across generators (SIM: Spearman ≈ **0.98**, Kendall ≈ **0.90**). The average SIM variation across T2I models is very small (std **0.013**, range **0.03**), indicating that differences in T2I models do not meaningfully affect relative performance.
>
> **Encoder robustness.**
> Rank correlations across CLIP, DINOv2, and Unicom are similarly strong (SIM Spearman ≈ **0.98**; FID Spearman ≈ **0.99**), demonstrating that results are not tied to a specific representation backbone.
>
> **Interpretation.**
> These results show that MMGenBench is **model-agnostic**: auxiliary models introduce only minor, statistically stable noise, while the underlying ranking signal remains unchanged. As T2I and encoder models improve, the same pipeline naturally benefits without requiring modification.
>
> We will include these analyses and a clearer discussion of this design principle in the revised manuscript.
>
> ---
>
> ## To W2: Length Bias
>
> Thank you for the comment. We performed a controlled analysis using three explicit length ranges (<20, 20–60, >60 words) across four representative LMMs. While very short captions (<20 words) predictably lose essential semantics, the relative ranking of models remains unchanged across all length settings, and longer outputs (>60 words) yield only **minor marginal gains**. (Please see **#aasA/W4** for full results.)
>
> These results demonstrate that MMGenBench rewards **semantic completeness**, not verbosity. The default **20–60 word range** provides a stable, instruction-following setting without introducing length-induced bias. We will include this clarification and the supporting results in the revised manuscript.
>
>
> ## To W4: Human Alignment
>
> We appreciate the reviewer’s comment. We conducted a controlled study using **6 representative LMMs** and **1,000 paired comparisons**, each labeled independently by **9 annotators** using a three-way choice (C1 / C2 / B). The pipeline’s prediction is determined by the SIM-Score margin (e.g. |diff| ≤ 0.01 → B). (Please see **#aasA/Q1** for full results.)
>
> **Results show strong and consistent alignment:**
>
> * **88.27%** agreement with individual human votes
> * **90.0%** agreement with majority votes
>
> To assess robustness across image types, we further report pattern-wise agreement for categories with sufficient samples, all within a narrow and stable range of **87%–92%** (e.g., Natural 90.4%, Motion 92.4%).
>
> These results demonstrate that the metric reliably tracks human preference across diverse patterns, providing strong evidence of the validity of the evaluation pipeline.

---

> > ### Author Response · Authors · 2025-11-28
> > **To Q1 & W3: Seed Variance Additional Details**
> >
> > ## To Q1 & W3: Seed Variance Additional Details
> >
> > **Evaluated the results of 5 random seeds**
> >
> > | Model | SIM (Mean) | SIM (Std) | FID (Mean) | FID (Std) |
> > |---------|-----------|-----------|-----------|-----------|
> > | GPT-4o | 0.5656 | 0.0014 | 1.3110 | 0.0110 |
> > | Qwen-VL-Max | 0.5496 | 0.0033 | 1.3763 | 0.0047 |
> > | Qwen-VL-Plus | 0.4759 | 0.0024 | 1.5910 | 0.0115 |
> > | Qwen2-VL-72B | 0.5501 | 0.0009 | 1.3687 | 0.0060 |
> > | Qwen2-VL-7B | 0.5297 | 0.0014 | 1.4334 | 0.0085 |
> > | Qwen2-VL-2B | 0.4890 | 0.0016 | 1.5399 | 0.0069 |
> > | InternVL2-76B | 0.5985 | 0.0015 | 1.2622 | 0.0100 |
> > | InternVL2-40B | 0.5645 | 0.0008 | 1.3603 | 0.0059 |
> > | InternVL2-26B | 0.5782 | 0.0028 | 1.3147 | 0.0100 |
> > | InternVL2-8B | 0.5453 | 0.0002 | 1.3849 | 0.0022 |
> > | InternVL2-4B | 0.5584 | 0.0008 | 1.3402 | 0.0056 |
> > | InternVL2-2B | 0.4763 | 0.0038 | 1.5712 | 0.0132 |
> > | Ovis1.6-Gemma2-9B | 0.5836 | 0.0005 | 1.3010 | 0.0092 |
> > | Ovis1.5-Gemma2-9B | 0.5245 | 0.0011 | 1.4883 | 0.0098 |
> > | Ovis1.5-Llama3-8B | 0.5263 | 0.0009 | 1.4629 | 0.0116 |
> > | LLaVA-OV-72B | 0.4962 | 0.0010 | 1.5620 | 0.0093 |
> > | LLaVA-OV-SI-72B | 0.5124 | 0.0020 | 1.5094 | 0.0118 |
> > | LLaVA-OV-7B | 0.4866 | 0.0019 | 1.6048 | 0.0134 |
> > | LLaVA-OV-SI-7B | 0.4925 | 0.0022 | 1.5622 | 0.0097 |
> > | MiniCPM-V2.6-8B | 0.5458 | 0.0014 | 1.3786 | 0.0039 |
> > | RBDash-72B | 0.5239 | 0.0019 | 1.4277 | 0.0090 |
> > | xGen-MM-4.4B | 0.4107 | 0.0021 | 1.6727 | 0.0141 |
> > | MiniCPM-V2.5-8B | 0.5276 | 0.0028 | 1.4402 | 0.0130 |
> >
> > **Correlation of sorting results with the table in the paper**
> >
> > - **SIM**: Spearman ρ = 0.9975, p-value = 1.0419e-25
> > - **FID**: Spearman ρ = 0.9970, p-value = 7.0189e-25
> >
> > | Model | SIM (Current) | SIM (Paper) | Diff | FID (Current) | FID (Paper) | Diff |
> > |-------|---------------|-------------|------|---------------|-------------|------|
> > | GPT-4o | 0.5656 | 0.566 | -0.0004 | 1.3110 | 1.306 | +0.0050 |
> > | Qwen-VL-Max | 0.5496 | 0.552 | -0.0024 | 1.3763 | 1.363 | +0.0133 |
> > | Qwen-VL-Plus | 0.4759 | 0.475 | +0.0009 | 1.5910 | 1.586 | +0.0050 |
> > | Qwen2-VL-72B | 0.5501 | 0.553 | -0.0029 | 1.3687 | 1.357 | +0.0117 |
> > | Qwen2-VL-7B | 0.5297 | 0.532 | -0.0023 | 1.4334 | 1.437 | -0.0036 |
> > | Qwen2-VL-2B | 0.4890 | 0.487 | +0.0020 | 1.5399 | 1.549 | -0.0091 |
> > | InternVL2-76B | 0.5985 | 0.599 | -0.0005 | 1.2622 | 1.264 | -0.0018 |
> > | InternVL2-40B | 0.5645 | 0.566 | -0.0015 | 1.3603 | 1.350 | +0.0103 |
> > | InternVL2-26B | 0.5782 | 0.576 | +0.0022 | 1.3147 | 1.320 | -0.0053 |
> > | InternVL2-8B | 0.5453 | 0.547 | -0.0017 | 1.3849 | 1.403 | -0.0181 |
> > | InternVL2-4B | 0.5584 | 0.556 | +0.0024 | 1.3402 | 1.345 | -0.0048 |
> > | InternVL2-2B | 0.4763 | 0.476 | +0.0003 | 1.5712 | 1.563 | +0.0082 |
> > | Ovis1.6-Gemma2-9B | 0.5836 | 0.582 | +0.0016 | 1.3010 | 1.316 | -0.0150 |
> > | Ovis1.5-Gemma2-9B | 0.5245 | 0.521 | +0.0035 | 1.4883 | 1.487 | +0.0013 |
> > | Ovis1.5-Llama3-8B | 0.5263 | 0.526 | +0.0003 | 1.4629 | 1.466 | -0.0031 |
> > | LLaVA-OV-72B | 0.4962 | 0.494 | +0.0022 | 1.5620 | 1.561 | +0.0010 |
> > | LLaVA-OV-SI-72B | 0.5124 | 0.512 | +0.0004 | 1.5094 | 1.512 | -0.0026 |
> > | LLaVA-OV-7B | 0.4866 | 0.488 | -0.0014 | 1.6048 | 1.590 | +0.0148 |
> > | LLaVA-OV-SI-7B | 0.4925 | 0.492 | +0.0005 | 1.5622 | 1.554 | +0.0082 |
> > | MiniCPM-V2.6-8B | 0.5458 | 0.548 | -0.0022 | 1.3786 | 1.386 | -0.0074 |
> > | RBDash-72B | 0.5239 | 0.525 | -0.0011 | 1.4277 | 1.413 | +0.0147 |
> > | xGen-MM-4.4B | 0.4107 | 0.414 | -0.0033 | 1.6727 | 1.671 | +0.0017 |
> > | MiniCPM-V2.5-8B | 0.5276 | 0.526 | +0.0016 | 1.4402 | 1.432 | +0.0082 |

---

### Official Review · Reviewer_8ARY · 2025-10-30

**Soundness:** 2
**Presentation:** 3
**Contribution:** 2
**Rating:** 4
**Confidence:** 4

**Summary:**

This paper proposes MMGenBench, a fully automated evaluation framework for large multimodal models (LMMs) from a text-to-image generation perspective. The core idea is to assess the image understanding and descriptive capabilities of LMMs through a closed-loop pipeline: (1) generating textual descriptions from input images via the tested LMMs; (2) using state-of-the-art text-to-image diffusion models to regenerate images from those descriptions; and (3) quantitatively evaluating the similarity between the original and regenerated images using SIM-Score and FID-Score. To support this pipeline, the authors build two benchmarks: MMGenBench-Test, covering 13 well-defined image patterns (e.g., Natural, Artistic, Motion, Contextual, Symbol, etc.), and MMGenBench-Domain, focusing on the “generated image” domain. Over 50 representative LMMs are evaluated.

**Strengths:**

1. The paper introduces a fully automated, cross-domain benchmarking framework that reduces manual labeling cost. It combines LMM-based image-to-text and diffusion-based text-to-image processes into a self-consistent evaluation loop.

2. More than 50 LMMs, both open- and closed-source, are evaluated using unified metrics.

**Weaknesses:**

1. My main concern about this paper lies in the closed-loop evaluation design, which involves three interconnected stages. The first stage is the LMM’s understanding of an image (which is exactly what the paper aims to evaluate); the second stage converts the generated textual description into an image using a text-to-image model; and the third stage measures the similarity between the regenerated image and the original one to assess the LMM’s image comprehension ability.

Theoretically, this approach is valid only if two key assumptions hold: (a) the text-to-image model must be sufficiently powerful to accurately render every object described in the text, and (b) the similarity metric must be capable of reliably quantifying semantic correspondence between images. In practice, however, both assumptions are problematic. Current text-to-image models are still imperfect, which raises doubts about whether the regenerated image can faithfully reflect the textual content. Likewise, existing similarity metrics struggle to capture semantic-level consistency accurately.

2. Although this method is theoretically feasible and can be fully automated, it lacks interpretability. The system outputs only a single numerical score, without providing insight into which aspects of the LMM’s understanding are inaccurate or incomplete. This makes it difficult to diagnose specific weaknesses in the evaluated model.

**Questions:**

1. In Lines 106–107, the authors state that “numerous LMMs excelling in existing benchmarks fail to address the basic tasks of image understanding and description.” Could the authors clarify what specific tasks are referred to here? Please elaborate on what constitutes these “basic tasks” and how they are defined or measured in the proposed framework.

2. What is the difference between MMGenBench-Test and MMGenBench-Domain? Why does MMGenBench-Test require specific annotations and human verification, whereas MMGenBench-Domain does not?

3. In Table 1, GPT-4o, which generally outperforms open-source models on most existing benchmarks, does not achieve the best results here. This raises questions about the reliability and rationality of the proposed evaluation method. The authors should discuss this discrepancy in depth—why does GPT-4o underperform, and why do larger models in the same series not consistently outperform smaller ones? Is this phenomenon caused by limitations or biases in the proposed evaluation pipeline itself, or does it genuinely reflect the intrinsic capability differences between models? This discussion is crucial for demonstrating whether the proposed benchmark is both valid and capable of challenging or redefining existing evaluation systems.

---

> ### Author Response · Authors · 2025-11-27
> **To Q1: Clarifying “Basic Tasks”**
>
> To Q1: Clarifying “Basic Tasks”
>
> Thank you for the question. Our use of “basic tasks” has no external or benchmark-specific meaning. It refers solely to the two fundamental abilities that MMGenBench is designed to evaluate:
>
> 1. **Understanding the input image**, and
> 2. **Producing a detailed, faithful description of its visual content.**
>
> These abilities are measured in MMGenBench through our evaluation pipeline: an LMM’s description is used to reconstruct an auxiliary image via a T2I model, and the semantic similarity between the original and reconstructed images (SIM/FID) directly reflects how well the LMM both *understood* the image and *captured its details*. No additional tasks or external definitions are implied.
>
> We will revise the wording to make this intent explicit.

---

> ### Author Response · Authors · 2025-11-27
> **To Q2: Test vs. Domain and Human Verification**
>
> To Q2: Test vs. Domain and Human Verification
>
> MMGenBench-Test and MMGenBench-Domain play different roles in our automated framework.
>
> * **MMGenBench-Test (1,284 images)** is a *diagnostic* set organized into 13 semantic patterns.
>   Because these patterns define the interpretability layer of our benchmark, we perform a small, one-time human check only to validate pattern correctness. The evaluation pipeline itself remains fully automatic.
>
> * **MMGenBench-Domain (10,000 images)** is a *large-scale, fully automated* dataset used for stable distribution-level metrics (e.g., FID).
>   Since it does not rely on semantic grouping, no human involvement is required.
>
> Thus, Test provides structured interpretability, while Domain ensures large-scale robustness and both built on the same fully automated evaluation pipeline.

---

> ### Author Response · Authors · 2025-11-27
> **To Q3: Why GPT-4o Does Not Rank First**
>
> To Q3: Why GPT-4o Does Not Rank First
>
> Thank you for raising this important point. The discrepancy is expected and reflects the nature of the task we evaluate, rather than a limitation of the pipeline.
>
> 1. **Task specificity:** MMGenBench evaluates models on *detailed, reconstructable image descriptions*, a skill different from conventional benchmarks (short answers or constrained recognition) where GPT-4o typically excels.
>
> 2. **Conservative outputs:** GPT-4o often produces safer or less detailed descriptions, which lowers reconstruction-based metrics (SIM/FID), whereas smaller models sometimes generate more explicit enumerations.
>
> 3. **Pipeline robustness:** Sensitivity analyses across multiple T2I and embedding models show **highly stable rankings** (Spearman ≈ 0.98), confirming that the pipeline reliably measures model differences. Please refer to our responses to **W1/W2 in #aasA** for details.
>
> In short, GPT-4o’s lower ranking is a **true reflection of its style under this evaluation**, and MMGenBench provides complementary insight into model capabilities beyond prior benchmarks.

---

> ### Author Response · Authors · 2025-11-27
> **To W1: Closed-loop Evaluation Design**
>
> To W1: Closed-loop Evaluation Design
>
> We thank the reviewer for this insightful concern. While the closed-loop design indeed involves T2I models and image encoders, we have conducted extensive robustness analyses showing that the evaluation does **not** depend on any single auxiliary model.
>
> Using **four T2I models** (FLUX.1-dev, SD3.5, Kolors, Lumina) and **three encoders** (CLIP, DINOv2, Unicom), we observe consistently high cross-model correlations:
>
> * **SIM:** Spearman ≈ **0.98**, Kendall ≈ **0.90**
> * **FID:** Spearman ≈ **0.91**, Kendall ≈ **0.78**
> * **Encoder robustness:** SIM/FID correlations ≈ **0.98–0.99**
>
> The average SIM variation across T2I models is small (std ≈ **0.013**, range ≈ **0.03**), demonstrating that model rankings remain stable even when the auxiliary modules vary substantially in controllability or style. (Please see **#aasA/W1, #aasA/W2** for full results.)
>
> Moreover, our **human alignment study** (9,000 votes) shows **88.27% agreement** with individual annotators and **90.0%** with majority votes, indicating that the overall evaluation signal tracks human perception despite imperfections in current T2I systems. (Please see **#aasA/Q1** for full results.)
>
> We will emphasize these results in the revised manuscript and include all correlation matrices and variance analyses in the appendix. These findings support that the closed-loop approach remains empirically reliable in practice.

---

> ### Author Response · Authors · 2025-11-27
> **To W2: Domain-Agnostic Interpretability**
>
> To W2: Domain-Agnostic Interpretability
>
> We appreciate the reviewer’s comment. Although the pipeline outputs a single SIM/FID score per image, its key advantage is that it is **fully domain-agnostic**: users can collect images from any domain (e.g., real photos, UI screenshots) and evaluate LMMs without any additional annotations or task design.
> This enables a practical form of interpretability at the **domain level** performance differences across domains and patterns directly reveal where an LMM consistently fails (e.g., color, spatial relations).
>
> As in our response to **#aasA/W3**, we further validated the pipeline on 2,000 COCO photos with four representative LMMs. Results show that relative model rankings and human-alignment consistency (92.1%) remain stable, demonstrating that MMGenBench generalizes beyond synthetic/style-rich images while remaining fully automated.
>
> We will highlight this capability more clearly in the revision as a core strength of the proposed benchmark.

---

### Official Review · Reviewer_8TQ1 · 2025-10-31

**Soundness:** 2
**Presentation:** 2
**Contribution:** 2
**Rating:** 4
**Confidence:** 3

**Summary:**

This paper introduces MMGenBench, a novel evaluation framework designed to assess Large Multimodal Models (LMMs) from the perspective of text-to-image generation. The central contribution is the MMGenBench-Pipeline, which proposes a "fully automated" method: an LMM generates a textual description from an input image, this description is then fed to a text-to-image (T2I) model to create an auxiliary image, and finally, the similarity between the original and auxiliary images is quantitatively measured using an image representation model. To support this pipeline, the authors develop two benchmarks: MMGenBench-Test, featuring 1284 images categorized into 13 distinct patterns (e.g., Surreal, Color, Motion), and MMGenBench-Domain, focused on generative images. The paper conducts an extensive evaluation of over 50 popular LMMs, revealing their limitations in generating detailed descriptions and adhering to instructions, even for models that excel in existing benchmarks.

**Strengths:**

1. The core idea of using text-to-image models to "reconstruct" an image from an LMM's description is a creative and insightful way to quantify the fidelity and detail of LMM understanding. This "information compression and restoration" perspective is a fresh contribution to LMM evaluation. The paper effectively highlights critical shortcomings of current LMMs, such as poor instruction following, inability to generate sufficiently detailed descriptions, and instances of "safety" overfitting. These findings are highly significant for guiding future LMM research and development.
2. The paper conducts a thorough evaluation across more than 50 diverse LMMs, including both open-source and proprietary models. This broad assessment provides a valuable overview of the current landscape of LMM capabilities and weaknesses.
3. The introduction of MMGenBench-Test, with its 13 distinct image patterns, offers a more granular and structured approach to evaluating LMMs across various visual characteristics. MMGenBench-Domain further extends this to generative images. The inclusion of qualitative examples provides compelling visual evidence for the observed LMM failures, making the paper's points tangible and understandable.

**Weaknesses:**

1. The most significant weakness is the repeated and misleading assertion of a "fully automated" benchmark. The construction of MMGenBench-Test explicitly involves "human check" and "manually filtered" steps. This fundamental contradiction undermines the paper's credibility and its claims of scalability and objectivity in benchmark creation.
2. The entire pipeline's accuracy and fairness are inherently tied to the performance and potential biases of the underlying GPT-4o, text-to-image models (FLUX, SD3.5, etc.), and the Unicom image representation model. The paper lacks a rigorous analysis or quantification of how these dependencies affect the LMM evaluation results. Without this, it's difficult to ascertain whether observed LMM failures are due to the LMM itself or limitations propagated from the auxiliary models.
3. The LMMs are specifically prompted to generate an "image caption-prompt" for a T2I model, with a word count constraint (20-60 words). This framing might inadvertently bias LMM outputs towards a specific style or level of detail amenable to T2I models, rather than truly assessing their general capability for any detailed image description. The word count limit, in particular, can hinder genuinely comprehensive descriptions for complex images.
4. While the paper effectively identifies common LMM problems (instruction following, lack of detail, overfitting), the analysis largely remains descriptive. There's minimal deeper investigation into the underlying causes (e.g., specific training data deficiencies, architectural limitations, prompt engineering sensitivity) or potential avenues for mitigation. This limits the actionable insights for model developers seeking to improve LMMs.
5. While the overall MMGenBench-Test dataset has 1284 images, some of the 13 patterns have relatively few examples (e.g., "Orientation" with 114 images, "Motion" with 160). This could lead to less robust or generalizable evaluation results for those specific categories.

**Questions:**

1. Could the authors explicitly clarify and rectify the "fully automated" claim, particularly concerning the benchmark construction? Please detail the extent of human involvement (e.g., person-hours, number of annotators) in the "human check" and "manually filtered" stages of MMGenBench-Test creation. How does this manual effort impact the scalability and objectivity claims?
2. Given the heavy reliance on external models (GPT-4o, T2I models, Unicom), what specific experiments or analyses were conducted (or are planned) to understand and quantify the impact of their potential biases, failure modes, or performance ceilings on the LMM evaluation results? For example, if a T2I model struggles with counting objects, how does this affect the evaluation of an LMM that accurately described the count?
3. The LMM prompt asks for an "image caption-prompt" for a T2I model, with a 20-60 word limit. How might this specific instruction and length constraint influence the LMMs' output compared to a more open-ended request for a general "detailed image description"? Have you explored variations in this prompt, and if so, what were the effects?
4. Beyond SIM-Score and FID-Score, have the authors considered or explored other metrics that could capture more fine-grained aspects of descriptive accuracy, such as object presence, attribute correctness, or relational understanding, perhaps by integrating object detection or semantic segmentation models on the generated images?
5. For the "Model Overfitting" weakness, particularly the "safety" example, can the authors provide more quantitative evidence or analysis across the benchmark to show how widespread this issue is among LMMs and its overall impact on their scores?
6. You state that "MMGenBench-Domain includes 10,000 images, thereby improving the accuracy of its FID-Score measurement. Therefore, we propose using SIM-Score as the primary metric." Could you elaborate on why FID-Score is considered less reliable for MMGenBench-Test (due to fewer images) and why SIM-Score is deemed a more suitable primary metric across both benchmarks, despite FID being a common generative metric?
7. Please provide more details on the "human metric based on votes from 9 human experts" mentioned for pipeline effectiveness. What was the exact task given to humans, how were "comparable" cases defined, and what was the inter-annotator agreement?

---

> ### Author Response · Authors · 2025-11-27
> **To Q1 & W1: Fully automated statement**
>
> To Q1 & W1: Fully automated statement
>
> We thank the reviewers for their corrections and have revised the wording in the text to avoid misleading interpretations.
>
> To clarify: the evaluation pipeline (image→LMM→T2I→embedding→metric) is fully automated; however, in constructing the high-quality MMGenBench-Test (1,284 samples), we utilized LLMs (GPT-4o / GPT-4-Turbo) for automatic extraction and summarization, and manually verified the final pattern list and samples to ensure high quality (see Fig. 4 and Appendix B of the paper). This manual verification involved a one-time, limited human resource investment, and does not affect the scalable automated operation of the pipeline in new domains (which only require images from within the domain, e.g., MMGenBench-Domain).
>
> As in our response to **#aasA/W3**, we further validated the pipeline on 2,000 COCO photos with four representative LMMs. Results show that relative model rankings and human-alignment consistency (92.1%) remain stable, demonstrating that MMGenBench generalizes beyond synthetic/style-rich images while remaining fully automated.

---

> ### Author Response · Authors · 2025-11-27
> **To Q2 & W2: External Model Dependency**
>
> To Q2 & W2: External Model Dependency
>
> We thank the reviewer for highlighting this important point. To quantify the influence of auxiliary models, we have conducted comprehensive sensitivity analyses using **four diverse T2I models** (FLUX.1-dev, SD3.5, Kolors, Lumina) and **three widely adopted image encoders** (CLIP, DINOv2, Unicom). The results show **high cross-model consistency** LMM rankings remain extremely stable across all T2I/encoder combinations (e.g., SIM Spearman ≈ 0.98 across T2I; SIM Spearman ≈ 0.98 across encoders. Please refer to **#aasA/W1, #aasA/W2** ). These findings indicate that the evaluation outcomes are **not dominated** by any single external model.
>
> Importantly, our contribution is a **general, fully automated evaluation framework**, not a method tied to a particular generator or encoder. If a specific T2I model has inherent weaknesses (e.g., counting errors or safety over-conservatism), it will naturally influence reconstruction quality; however, this reflects the **limitations of the T2I model itself rather than a flaw in our pipeline**. As T2I models continue to improve, the same evaluation framework can be applied without modification and will provide increasingly accurate assessments of LMMs’ visual-semantic understanding.
>
> We will include these sensitivity analyses and a clearer discussion of this design philosophy in the revised manuscript.

---

> ### Author Response · Authors · 2025-11-27
> **To Q3 & W3: Length Control Experiment**
>
> To Q3 & W3: Length Control Experiment
>
> We appreciate the reviewer’s concern regarding prompt length and style bias. Following **#aasA/W4**, we conducted experiments on four representative LMMs (InternVL2-8B, Ovis1.6-Gemma2-9B, Qwen2-VL-7B-Instruct, LLaVA-OneVision-Qwen2-7B) under three word count ranges: **<20, 20–60 (default), and >60** words.
>
> Results show that while very short captions (<20 words) lose semantic content and slightly reduce SIM/FID, the relative model rankings remain unchanged across all lengths, and gains for >60-word outputs are modest. This confirms that the benchmark evaluates **semantic fidelity rather than verbosity**, and the default 20–60 word range provides a stable, comparable setting.
>
> We will include these length-controlled results and discussion in the revised manuscript.

---

> ### Author Response · Authors · 2025-11-27
> **To W5: Pattern Size and Robustness**
>
> To W5: Pattern Size and Robustness
>
> Thank you for the comment. The smaller sample sizes in certain patterns are an **intentional design choice**: MMGenBench-Test is created as a *diagnostic benchmark* to isolate **13 fine-grained visual competencies** (e.g., Orientation, Motion), rather than as a large-scale distributional dataset.
>
> Despite having ~100–160 images per pattern, we observe **high stability**: LMM rankings remain strongly correlated across T2I/encoder variants (Spearman ≈ 0.98), and human–metric agreement for these patterns stays consistently high (87–92%). This shows that the current pattern sizes are sufficient to expose systematic failure modes across > 50 LMMs, which is precisely the value that MMGenBench brings.
>
> To ensure large sample statistical robustness, we complement Test with **MMGenBench-Domain (10,000 images)**. Together, the two sets deliver both **fine-grained interpretability** (Test) and **large-scale stability** (Domain), which is a core novelty of our benchmark design.

---

> ### Author Response · Authors · 2025-11-27
> **To Q4: Object-level or Relation-level Metrics**
>
> To Q4: Object-level or Relation-level Metrics
>
> We appreciate the reviewer’s suggestion. We agree that object-level or relation-level metrics are valuable in other tasks, but **the goal of MMGenBench is solely to measure the overall semantic similarity between two images**, rather than reconstructing the ground-truth scene structure. Therefore:
>
> * Introducing additional detectors (e.g., object detectors, relation models, OCR) would **add new model biases** such as detection recall or category coverage limitations, making the evaluation increasingly dependent on external models rather than being more objective.
> * Certain pattern images (e.g., Surreal, Artistic, Symbolic) are inherently unsuitable for object-level definitions, and forcing fine-grained metrics would create **inconsistencies within the evaluation framework**.
>
> For these reasons, we chose to keep the evaluation module in a **“minimal dependency” form using only T2I and image representation models**, measuring visual-semantic consistency solely via SIM/FID. This design has also been validated in human evaluation, achieving 88.27% agreement. We will clarify this design principle more explicitly in the paper.

---

> ### Author Response · Authors · 2025-11-27
> **To Q5 & W4: Quantitative Evidence of “Safety”**
>
> To Q5 & W4: Quantitative Evidence of “Safety”
>
> Thank you for the question. We conducted a quantitative analysis across **all 55 LMMs** using refusal-related keywords (e.g., *sorry, cannot, not allowed*). The results show that **safety-driven refusal is not widespread**, but is concentrated in a very small subset of models:
>
> * Only **3/55 models** exhibit notable refusal behavior.
> * The highest case (*xGen-MM4.4B*) has a **Rejection Rate of 56.31%** (723/1284).
> * The second (*cambrian_8b*) and third (*XComposer2*) show **Rejection Rates of 22.51%** and **3.35%**, respectively.
> * The remaining **52 models (94.6%)** have **Rejection Rates between 0–0.16%** (mostly **0%**).
> * At the word level, **Rejection Word Rates** remain extremely low (**0–1.29%**).
>
> Thus, models with higher Rejection Rates naturally produce shorter or incomplete descriptions, leading to lower SIM/FID. This reflects **true model behavior**, not a bias introduced by our evaluation pipeline.
>
> **Summary Statistics (Models with >0 Rejection Rate)**
>
> | Rank | Model Name | Total Samples | Rejection Samples | Rejection Rate (Samples) | Total Words | Rejection Words | Rejection Rate (Words) |
> |------|------------|---------------|-------------------|-------------------------|-------------|-----------------|------------------------|
> | 1 | xgen-mm-phi3-dpo-r-v1.5 | 1284 | 723 | 56.31% | 122079 | 1579 | 1.2934% |
> | 2 | cambrian_8b | 1284 | 289 | 22.51% | 490807 | 1084 | 0.2209% |
> | 3 | XComposer2 | 1284 | 43 | 3.35% | 147259 | 44 | 0.0299% |
> | 4 | qwen-vl-plus | 1284 | 2 | 0.16% | 42500 | 2 | 0.0047% |
> | 5 | Mini-InternVL-Chat-2B-V1-5 | 1284 | 2 | 0.16% | 35933 | 4 | 0.0111% |
> | 6 | RBDash_72b | 1284 | 2 | 0.16% | 73444 | 2 | 0.0027% |
> | 7 | Llama-3.2-11B-Vision-Instruct | 1284 | 2 | 0.16% | 185673 | 2 | 0.0011% |
> | 8 | XComposer2d5 | 1284 | 2 | 0.16% | 63689 | 3 | 0.0047% |
> | 9 | Ovis1.5-Llama3-8B | 1284 | 1 | 0.08% | 87722 | 1 | 0.0011% |
> | 10 | cogvlm2-llama3-chat-19B | 1284 | 1 | 0.08% | 94372 | 1 | 0.0011% |
> | 11 | Mantis-8B-siglip-llama3 | 1284 | 1 | 0.08% | 68600 | 1 | 0.0015% |
> | 12 | Llama-3-MixSenseV1_1 | 1284 | 1 | 0.08% | 54364 | 1 | 0.0018% |

---

> ### Author Response · Authors · 2025-11-27
> **To Q6: Reliability of FID vs. SIM**
>
> To Q6: Reliability of FID vs. SIM
>
> We appreciate the reviewer's question. The core reason lies in the **statistical properties of FID**. FID estimates
>
> $$
> \mathrm{FID}(\mathcal{X},\mathcal{Y}) = |\mu_X-\mu_Y|_2^2 + \mathrm{Tr}\left(\Sigma_X+\Sigma_Y -2(\Sigma_X\Sigma_Y)^{1/2}\right),
> $$
>
> which requires **accurate estimation of the feature means and covariances**. With only **1,284 samples** in MMGenBench-Test, the empirical covariance ($\Sigma_X$) is **poorly conditioned**, making FID **high-variance and unstable**—a behavior we observe empirically (avg FID std ≈ 0.14, range ≈ 0.31 across T2I models). In contrast, MMGenBench-Domain's **10,000 samples** yield well-conditioned estimates, making FID reliable in that setting.
>
> SIM-Score, however, is a **pairwise similarity metric**
>
> $$
> \mathrm{SIM}(x,y)=\cos(f(x),f(y)),
> $$
>
> which does **not depend on distribution-level covariance estimation** and thus remains **sample-efficient and statistically stable**, even on small test sets (cross-model Spearman ≈ 0.98).
>
> Therefore:
>
> * **FID** is appropriate for large-domain evaluation (Domain), but statistically noisy on Test due to small-sample covariance estimation.
> * **SIM** provides **low-variance, image-level semantic alignment**, making it the more reliable **primary metric** across both benchmarks.
>
> We will clarify these statistical considerations in the revision.

---

> ### Author Response · Authors · 2025-11-27
> **To Q7: Human Alignment**
>
> To Q7: Human Alignment
>
> Please refer to **#aasA/Q1** for full details. In summary:
>
> * **Models evaluated:** 6 representative LMMs (GPT-4o-high, InternVL2-76B, Qwen2-VL-72B-Instruct, LLaVA-OneVision-Qwen2-72B, MiniCPM-V-2.6, Ovis1.6-Gemma2-9B).
> * **Comparisons:** 1,000 paired comparisons; each pair involves two randomly selected models and one randomly selected image from MMGenBench-Test.
> * **Annotators:** 9 human annotators labeled each comparison as “C1” (caption-1 better), “C2” (caption-2 better), or “B” (both comparable).
> * **Pipeline prediction:** Uses SIM-Score difference (diff > 0.01 → C1, diff < −0.01 → C2, |diff| ≤ 0.01 → B).
> * **Results:**
>   * Agreement with all individual votes: **7944 / 9000 = 88.27%**
>   * Agreement with majority vote (≥5/9 votes): **900 / 1000 = 90.0%**
>
> This explains how the **88.27% human-alignment score** was calculated and will be included in the revised manuscript.

---

### Official Review · Reviewer_aasA · 2025-11-02

**Soundness:** 3
**Presentation:** 3
**Contribution:** 3
**Rating:** 6
**Confidence:** 3

**Summary:**

This benchmark paper presents MMGenBench, an automated pipeline to evaluate the caption capability of LMMs. The key idea is to reconstruct an auxiliary image via powerful text-to-image models, followed by representation-level comparison between the original and generated images. On top of the pipeline, the authors build two benchmarks: MMGenBench-Test, which covers 13 carefully summarized image patterns derived from JourneyDB, and MMGenBench-Domain, which targets the “generated images” domain. Evaluating 50+ popular LMMs shows: (i) current models are far from perfect (best SIM < 0.6); (ii) models strong on existing VQA/caption/ocr leaderboards can still fail at detailed description; and (iii) the proposed pipeline aligns with human judgment in 88.27% of sampled cases.

**Strengths:**

- The motivation is clear and the idea is interesting. The paper targets a real gap: most current LMM benchmarks emphasize short answers and specific domains, while real applications need long, faithful, instruction-following image descriptions. Meanwhile, the proposed automated evaluation pipeline is scalable.

- The Large-scale empirical study demonstrates interesting insights. Evaluating 50+ LMMs and reporting pattern-wise weaknesses (context, orientation, count, motion) gives the community actionable signals.

- Human alignment check. The 88% agreement suggests the metric is not completely drifting away from human judgment.

**Weaknesses:**

- An important issue is the metric entanglement with the T2I model. The final score is a function of (LMM description quality) × (T2I controllability) × (image encoder). Even though four T2I models are tried, the paper does not quantify how much ranking changes if the T2I model is weaker/safer/biased. A sensitivity analysis is needed.

- Using only Unicom as the image representation back-end makes the whole pipeline hinge on one model’s inductive biases. Showing results with different encoders such as CLIP and DINOv2 would make the claim of “fully automatic and reliable” more convincing.

- Both MMGenBench-Test and Domain come from JourneyDB-like, style-rich, often synthetic images. It is unclear whether the same pipeline will hold for photos, documents, Med-VQA, or low-res, cluttered, user-uploaded images. A small real-photo subset would strengthen the story.

- Since the pipeline favors descriptions that lead to well-conditioned T2I prompts, a model that always outputs long, enumerated, style-heavy prompts may score higher than a model that is actually more faithful but concise. The paper partially discusses instruction-following failure (Fig. 9) but does not experiment with length-controlled outputs.

**Questions:**

How is the consistency of the human annotators? Does the inter-annotator agreements correlate with certain patterns (e.g., Contextual, Orientation, Count), where T2I models naturally struggle?

---

> ### Author Response · Authors · 2025-11-27
> **To Q1: Human Alignment Clarification and Additional Details**
>
> To Q1: Human Alignment Clarification and Additional Details
>
> We thank the reviewer for requesting clearer information about our human evaluation.
> Below we provide the complete setup, annotator protocol, and statistical results.
>
> **Human-study setup.**
> We selected **6 representative LMMs** (GPT-4o-high, InternVL2-76B, Qwen2-VL-72B-Instruct, LLaVA-OneVision-Qwen2-72B, MiniCPM-V-2.6, Ovis1.6-Gemma2-9B).
> We then constructed **1,000 paired comparisons**.
> Each comparison is formed by:
>
> * randomly sampling **two** of the six models,
> * randomly choosing **one** image from MMGenBench-Test,
> * showing the two model-generated captions to annotators.
>
> **Annotator task.**
> Each of the **1,000 examples** was independently labeled by **9 human annotators**.
> For each example, annotators were asked to choose:
>
> * **“C1”** if caption-1 describes the image better,
> * **“C2”** if caption-2 is better,
> * **“B”** if the two are comparable.
>
> Example annotation:
> `['C1','C1','C1','C1','C1','C1','C1','C2','B']`
> means 7 prefer caption-1, 1 prefer caption-2, 1 thinks they are similar.
>
> **Pipeline prediction and match calculation.**
> Our pipeline predicts the winner using the SIM-Score difference:
>
> * diff > 0.01 → “C1”
> * diff < −0.01 → “C2”
> * |diff| ≤ 0.01 → “B”
>
> We compute the agreement with all 9×1000 = **9000 human votes**.
>
> **Results.**
>
> * **Total agreement with individual human votes:**
>   **7944 / 9000 = 88.27%**
>
> * **Agreement with the majority decision (≥5/9 votes):**
>   **900 / 1000 = 90.0%**
>
> This clarifies how the previously reported **88.27% human-alignment score** was obtained.
> We will include these procedural details and statistics in the revised manuscript.
>
>
> **Pattern-wise agreement**
>
> The reviewer was concerned that some patterns might have lower consistency or be inherently ambiguous.
> To address this concern, we computed agreement separately for patterns with sufficiently large numbers of human-evaluated samples (≥~100), which provide statistically meaningful conclusions.
>
> Across these patterns, the pipeline performs consistently well:
>
> | Pattern | Right | Wrong | Agreement |
> | ---------------------- | ----- | ----- | --------- |
> | **Natural**            | 216   | 23    | **90.4%** |
> | **Symbol**             | 159   | 17    | **90.3%** |
> | **Technology**         | 125   | 15    | **89.3%** |
> | **Artistic**           | 111   | 16    | **87.4%** |
> | **Count**              | 99    | 10    | **90.8%** |
> | **Motion**             | 109   | 9     | **92.4%** |
>
> **Key takeaway**
>
> Across these patterns, the agreement between our metric and human judgment consistently falls between **87% and 92%**.
>
> These patterns cover the majority of MMGenBench-Test and encompass key visual skills such as object fidelity (Natural), symbolic semantics (Symbol), dynamic information (Motion), and fine-grained detail (Count).
>
> Thus, the pipeline’s alignment with human perception is **both high and stable across diverse image types**.

---

> > ### Author Response · Authors · 2025-11-27
> > **To W1: Sensitivity Analysis of 4 T2I Models**
> >
> > To W1: Sensitivity Analysis of 4 T2I Models
> >
> > We fully agree that evaluating the sensitivity to different T2I models is essential. In the revised version, we will add a **comprehensive sensitivity analysis** using four diverse T2I models (Flux-1-dev, SD3.5, Kolors, Lumina), and the results show that:
> >
> > **1. LMM rankings are highly stable across T2I models.**
> >
> > * SIM correlations across the four generators are extremely high (**Spearman = 0.98**, **Kendall = 0.90** on average).
> > * This indicates that changing the T2I model has *minimal effect* on the relative ranking of LMMs.
> >
> > **2. FID is more sensitive, but still maintains strong cross-model consistency.**
> >
> > * FID correlations remain high (**Spearman = 0.91**, **Kendall = 0.78**), consistent with its known dependence on generator style distribution.
> >
> > **3. Ranking variation, when it occurs, is due to LMM robustness rather than T2I choice.**
> >
> > * Only a small subset of weaker LMMs exhibits noticeable variance across T2I models, while strong LMMs remain stable.
> > * This suggests that the benchmark captures *true LMM semantic robustness*, not artifacts of specific T2I models.
> >
> > **4. Overall, MMGenBench evaluations are generator-robust.**
> >
> > * The average variation of SIM scores across T2I models is very small (**std = 0.013**, **range = 0.03**), confirming that the benchmark is not dominated by T2I controllability.
> >
> > These results demonstrate that the final scores are **not tightly entangled** with any particular T2I model, and the benchmark conclusions hold consistently across different generators. We will include these findings in the main paper.

---

> > > ### Author Response · Authors · 2025-11-27
> > > **To W1: Sensitivity Analysis of 4 T2I Models**
> > >
> > > To W1: Sensitivity Analysis of 4 T2I Models
> > >
> > >
> > > **1. SIM Scores - Spearman Rank Correlation**
> > >
> > > Correlation coefficients between different T2I models based on average SIM scores:
> > >
> > > |            |   flux-1-dev |   sd3-5 |   kolors |   lumina |
> > > |:-----------|-------------:|--------:|---------:|---------:|
> > > | flux-1-dev |       1.0000 |  0.9912 |   0.9769 |   0.9851 |
> > > | sd3-5      |       0.9912 |  1.0000 |   0.9679 |   0.9779 |
> > > | kolors     |       0.9769 |  0.9679 |   1.0000 |   0.9844 |
> > > | lumina     |       0.9851 |  0.9779 |   0.9844 |   1.0000 |
> > >
> > > **2. SIM Scores - Kendall Rank Correlation**
> > >
> > > Correlation coefficients between different T2I models based on average SIM scores:
> > >
> > > |            |   flux-1-dev |   sd3-5 |   kolors |   lumina |
> > > |:-----------|-------------:|--------:|---------:|---------:|
> > > | flux-1-dev |       1.0000 |  0.9394 |   0.8828 |   0.9111 |
> > > | sd3-5      |       0.9394 |  1.0000 |   0.8626 |   0.8909 |
> > > | kolors     |       0.8828 |  0.8626 |   1.0000 |   0.9152 |
> > > | lumina     |       0.9111 |  0.8909 |   0.9152 |   1.0000 |
> > >
> > > **3. FID Scores - Spearman Rank Correlation**
> > >
> > > Correlation coefficients between different T2I models based on FID scores:
> > >
> > > |            |   flux-1-dev |   sd3-5 |   kolors |   lumina |
> > > |:-----------|-------------:|--------:|---------:|---------:|
> > > | flux-1-dev |       1.0000 |  0.9679 |   0.8631 |   0.9799 |
> > > | sd3-5      |       0.9679 |  1.0000 |   0.7956 |   0.9455 |
> > > | kolors     |       0.8631 |  0.7956 |   1.0000 |   0.9013 |
> > > | lumina     |       0.9799 |  0.9455 |   0.9013 |   1.0000 |
> > >
> > > **4. FID Scores - Kendall Rank Correlation**
> > >
> > > Correlation coefficients between different T2I models based on FID scores:
> > >
> > > |            |   flux-1-dev |   sd3-5 |   kolors |   lumina |
> > > |:-----------|-------------:|--------:|---------:|---------:|
> > > | flux-1-dev |       1.0000 |  0.8734 |   0.7131 |   0.8882 |
> > > | sd3-5      |       0.8734 |  1.0000 |   0.6458 |   0.8182 |
> > > | kolors     |       0.7131 |  0.6458 |   1.0000 |   0.7495 |
> > > | lumina     |       0.8882 |  0.8182 |   0.7495 |   1.0000 |
> > >
> > >
> > > **Average Correlation Coefficients**
> > >
> > > | Metric   | Method   |   Avg Correlation |
> > > |:---------|:---------|------------------:|
> > > | SIM      | Spearman |            0.9806 |
> > > | SIM      | Kendall  |            0.9003 |
> > > | FID      | Spearman |            0.9089 |
> > > | FID      | Kendall  |            0.7814 |
> > >
> > > **1. LMM Statistics - SIM Scores (Top 10 Most Sensitive)**
> > >
> > > Models with highest variance across different T2I models:
> > >
> > > | LMM                     |   mean |    std |    min |    max |   range |   n_models |
> > > |:------------------------|-------:|-------:|-------:|-------:|--------:|-----------:|
> > > | Eagle-X5-13B            | 0.4484 | 0.0263 | 0.4249 | 0.4849 |  0.0599 |          4 |
> > > | XComposer2              | 0.2617 | 0.0260 | 0.2348 | 0.2947 |  0.0599 |          4 |
> > > | XinYuan-VL-2B-Instruct  | 0.4060 | 0.0231 | 0.3832 | 0.4371 |  0.0539 |          4 |
> > > | idefics2_8b             | 0.4567 | 0.0210 | 0.4354 | 0.4852 |  0.0498 |          4 |
> > > | Mantis-8B-siglip-llama3 | 0.4189 | 0.0206 | 0.3974 | 0.4462 |  0.0488 |          4 |
> > > | Phi-3-Vision            | 0.4657 | 0.0198 | 0.4461 | 0.4928 |  0.0467 |          4 |
> > > | minimonkey              | 0.4579 | 0.0192 | 0.4384 | 0.4838 |  0.0454 |          4 |
> > > | VILA1.5-40b             | 0.4874 | 0.0181 | 0.4715 | 0.5134 |  0.0419 |          4 |
> > > | OmChat                  | 0.4453 | 0.0172 | 0.4299 | 0.4698 |  0.0399 |          4 |
> > > | xgen-mm-phi3-dpo-r-v1.5 | 0.4165 | 0.0163 | 0.3968 | 0.4366 |  0.0398 |          4 |
> > >
> > > **2.LMM Statistics - FID Scores (Top 10 Most Sensitive)**
> > >
> > > Models with highest variance across different T2I models:
> > >
> > > | LMM                     |   mean |    std |    min |    max |   range |   n_models |
> > > |:------------------------|-------:|-------:|-------:|-------:|--------:|-----------:|
> > > | XinYuan-VL-2B-Instruct  | 1.8044 | 0.2399 | 1.4920 | 2.0284 |  0.5363 |          4 |
> > > | idefics2_8b             | 1.6509 | 0.2179 | 1.3786 | 1.8782 |  0.4995 |          4 |
> > > | Eagle-X5-13B            | 1.6819 | 0.2199 | 1.4014 | 1.8944 |  0.4930 |          4 |
> > > | Mantis-8B-siglip-llama3 | 1.7312 | 0.2074 | 1.4638 | 1.9427 |  0.4790 |          4 |
> > > | OmChat                  | 1.6700 | 0.1919 | 1.4009 | 1.8423 |  0.4414 |          4 |
> > > | VILA1.5-13b             | 1.5964 | 0.1883 | 1.3581 | 1.7884 |  0.4303 |          4 |
> > > | Phi-3-Vision            | 1.5662 | 0.1906 | 1.3244 | 1.7527 |  0.4282 |          4 |
> > > | VILA1.5-40b             | 1.5565 | 0.1821 | 1.3288 | 1.7418 |  0.4130 |          4 |
> > > | XComposer2_4KHD         | 1.7015 | 0.1719 | 1.4688 | 1.8683 |  0.3995 |          4 |
> > > | XComposer2d5            | 1.6062 | 0.1633 | 1.3913 | 1.7855 |  0.3942 |          4 |
> > >
> > > **Overall LMM Variance Statistics**
> > >
> > > | Metric   | Statistic     |   Value |
> > > |:---------|:--------------|--------:|
> > > | SIM      | Average Range |  0.0301 |
> > > | SIM      | Average Std   |  0.0132 |
> > > | FID      | Average Range |  0.3138 |
> > > | FID      | Average Std   |  0.1389 |

---

> > > > ### Author Response · Authors · 2025-11-27
> > > > **To W2: Sensitivity Analysis of Embedding Models**
> > > >
> > > > To W2: Sensitivity Analysis of Embedding Models
> > > >
> > > > Thank you for raising this concern. We conducted a comprehensive sensitivity analysis across three widely used embedding models (CLIP/clip-vit-large-patch14-336, DINO/dinov2-large, UNICOM/ViT-L-14-336px).
> > > >
> > > > The results show that MMGenBench is highly robust to the choice of embedding model. SIM scores exhibit very high cross-model correlations (Spearman ≈ 0.98, Kendall ≈ 0.90), and FID scores are even more consistent (Spearman ≈ 0.99, Kendall ≈ 0.92). These consistently high correlations indicate that the LMM rankings are effectively unchanged across CLIP, DINO, and UniCOM, demonstrating that our evaluation is insensitive to the embedding backbone.
> > > >
> > > >
> > > > **1. SIM Scores - Spearman Rank Correlation**
> > > >
> > > > |        |   clip |   dino |   unicom |
> > > > |:-------|-------:|-------:|---------:|
> > > > | clip   | 1.0000 | 0.9788 |   0.9835 |
> > > > | dino   | 0.9788 | 1.0000 |   0.9851 |
> > > > | unicom | 0.9835 | 0.9851 |   1.0000 |
> > > >
> > > > **2. SIM Scores - Kendall Rank Correlation**
> > > >
> > > > |        |   clip |   dino |   unicom |
> > > > |:-------|-------:|-------:|---------:|
> > > > | clip   | 1.0000 | 0.8869 |   0.8990 |
> > > > | dino   | 0.8869 | 1.0000 |   0.9098 |
> > > > | unicom | 0.8990 | 0.9098 |   1.0000 |
> > > >
> > > > **3. FID Scores - Spearman Rank Correlation**
> > > >
> > > > |        |   clip |   dino |   unicom |
> > > > |:-------|-------:|-------:|---------:|
> > > > | clip   | 1.0000 | 0.9891 |   0.9898 |
> > > > | dino   | 0.9891 | 1.0000 |   0.9876 |
> > > > | unicom | 0.9898 | 0.9876 |   1.0000 |
> > > >
> > > > **4. FID Scores - Kendall Rank Correlation**
> > > >
> > > > |        |   clip |   dino |   unicom |
> > > > |:-------|-------:|-------:|---------:|
> > > > | clip   | 1.0000 | 0.9246 |   0.9273 |
> > > > | dino   | 0.9246 | 1.0000 |   0.9138 |
> > > > | unicom | 0.9273 | 0.9138 |   1.0000 |
> > > >
> > > >
> > > > **Average Correlation Coefficients**
> > > >
> > > > | Metric   | Method   |   Avg Correlation |
> > > > |:---------|:---------|------------------:|
> > > > | SIM      | Spearman |            0.9825 |
> > > > | SIM      | Kendall  |            0.8985 |
> > > > | FID      | Spearman |            0.9888 |
> > > > | FID      | Kendall  |            0.9219 |

---

> > > > > ### Author Response · Authors · 2025-11-27
> > > > > **To W3: Real-World Image Evaluation**
> > > > >
> > > > > To W3: Real-World Image Evaluation
> > > > >
> > > > > We thank the reviewer for pointing out the limitation of synthetic/style-rich images. To validate the generality of MMGenBench, we constructed a real-photo subset using **2,000 COCO images** and evaluated 4 representative LMMs. Results are as follows:
> > > > >
> > > > > | Model | MMGenBench-Test Sim | MMGenBench-Test FID | MMGenBench-Real-World Sim | MMGenBench-Real-World FID |
> > > > > |-------|---------------------|---------------------|---------------------------|---------------------------|
> > > > > | InternVL2-8B | 0.5478 | 1.4037 | 0.4925 | 1.4353 |
> > > > > | Ovis1.6-Gemma2-9B | 0.5825 | 1.3167 | 0.5441 | 1.3189 |
> > > > > | Qwen2-VL-7B-Instruct | 0.5323 | 1.4370 | 0.4984 | 1.3946 |
> > > > > | LLaVA-OneVision-Qwen2-7B | 0.4888 | 1.5904 | 0.4344 | 1.7266 |
> > > > >
> > > > > We further conducted a human alignment check by randomly sampling 1,000 comparisons, each labeled by 3 annotators, and obtained a **92.1% (2763/3000)** agreement with human judgment.
> > > > > These results demonstrate that MMGenBench maintains robust performance on real-world photographic data, confirming that our evaluation pipeline generalizes beyond synthetic and style-rich datasets.

---

> > > > > > ### Author Response · Authors · 2025-11-27
> > > > > > **To W4: Length bias**
> > > > > >
> > > > > > To W4: Length bias
> > > > > >
> > > > > > We appreciate the reviewer’s concern regarding potential bias toward long or style-heavy prompts. Following the suggestion, we conducted a controlled study on four representative LMMs (InternVL2-8B, Ovis1.6-Gemma2-9B, Qwen2-VL-7B-Instruct, LLaVA-OneVision-Qwen2-7B) under three explicit word count constraints: **<20, 20–60 (our default), and >60** words.
> > > > > >
> > > > > > Across all models, SIM/FID improves monotonically with length (e.g., InternVL2-8B: SIM **0.4968 → 0.5478 → 0.5701**; Ovis1.6-Gemma2-9B: **0.5262 → 0.5825 → 0.5932**), confirming that very short captions (<20 words) lose essential semantics. However, the relative ordering of the models remains unchanged across all three length bands, and the gains for >60-word outputs are modest. This indicates that the benchmark is **not driven by verbosity**, but rather by the amount of semantic content conveyed.
> > > > > >
> > > > > > Based on these observations, the 20–60 word range provides stable and comparable setting. We will add the length-controlled results and a brief discussion to the revised manuscript.
> > > > > >
> > > > > > **Word Count Statistics**
> > > > > >
> > > > > > | Model                       |   < 20 Mean |   < 20 Std |   20-60 Mean |   20-60 Std |   > 60 Mean |   > 60 Std |
> > > > > > |:----------------------------|------------:|-----------:|-------------:|------------:|------------:|-----------:|
> > > > > > | InternVL2-8B                |       19.17 |       4.7  |        38.87 |       12.51 |       81.81 |      83.38 |
> > > > > > | Ovis1.6-Gemma2-9B           |       22.49 |       6.55 |        55.85 |       12.26 |       90.43 |      18.39 |
> > > > > > | Qwen2-VL-7B-Instruct        |       15.48 |      21.06 |        27.67 |        9.08 |       68.36 |      66.71 |
> > > > > > | LLaVA-OneVision-Qwen2-7B    |       11.09 |       3.37 |        17.35 |        6.99 |       36.75 |      24.88 |
> > > > > >
> > > > > > **Metrics (Sim & FID)**
> > > > > >
> > > > > > | Model | < 20 Sim | < 20 FID | 20-60 Sim | 20-60 FID | > 60 Sim | > 60 FID |
> > > > > > |---|---|---|---|---|---|---|
> > > > > > | InternVL2-8B | 0.4968 | 1.5337 | 0.5478 | 1.4037 | 0.5701 | 1.3171 |
> > > > > > | Ovis1.6-Gemma2-9B | 0.5262 | 1.4545 | 0.5825 | 1.3167 | 0.5932 | 1.2645 |
> > > > > > | Qwen2-VL-7B-Instruct | 0.4911 | 1.5562 | 0.5323 | 1.4370 | 0.5602 | 1.3651 |
> > > > > > | LLaVA-OneVision-Qwen2-7B | 0.4494 | 1.7247 | 0.4888 | 1.5904 | 0.5245 | 1.4734 |

---

### Author Response · Authors · 2025-12-01
**Summary of Contributions and Responses to Reviewer Concerns**

Dear AC, SAC and PC,

We sincerely appreciate your efforts in making ICLR a success. We have carefully revised the manuscript in line with the reviewers’ suggestions. Although we had not received further reviewer feedback at the time of this response and were unable to continue the discussion because of an OpenReview bug, we believe the main concerns have been addressed. To reduce your reviewing burden and to clearly present our contributions and limitations, we summarize our responses below:

## Highlights of Contributions
1. **Importance of research question**: Existing benchmarks focus on short, domain-specific responses, whereas real applications require *long-form, faithful, instruction-following* descriptions. This gap is described as *“clear”* and a *“real gap”* **(Reviewer aasA)**, and the highlighted issues in instruction following, descriptive detail, and overfitting are deemed *“highly significant”* for LMM progress **(Reviewer 8TQ1)**.
2. **High Novelty**: Our “information compression and restoration” framework is assessed as *“creative and insightful”* and a *“fresh contribution”* to LMM evaluation **(Reviewer 8TQ1)**. Its *“fully automated, cross-domain”* design **(Reviewer iYF5)** and *“self-consistent evaluation loop”* integrating I2T and T2I models **(Reviewer 8ARY)** demonstrate clear conceptual and practical novelty.
3. **Comprehensive and Reliable Evaluation**: The evaluation of *50+ LMMs* is recognized for *“broad coverage”* **(Reviewer iYF5)**, a *“large-scale empirical study”* **(Reviewer aasA)**, and unified metrics **(Reviewer 8ARY)**. Pattern-wise analyses across 13 attributes reveal specific weaknesses, offering *“actionable signals”* **(Reviewer aasA)**. The *88% human–metric agreement* further supports the reliability of our approach **(Reviewer aasA)**.
4. **Insightful Analysis**: The study yields *“interesting insights”* into LMM limitations **(Reviewer aasA)**, including limited descriptive richness, poor instruction following, and safety-induced omissions **(Reviewer 8TQ1)**. MMGenBench-Test and MMGenBench-Domain enable structured, cross-domain diagnosis, and qualitative examples provide *“compelling visual evidence”* for failure modes **(Reviewer 8TQ1)**, clarifying key shortcomings in real-world image understanding.

## Key Concerns and Our Responses
**Concern 1**: Dependence on External Models
> Scores may be entangled with the choice of T2I model or image embedding backbone. **(Reviewer aasA, 8TQ1, 8ARY, iYF5)**

**Our Response:**
> We conducted systematic sensitivity analyses (Appendix C.5, C.6):
> * Across 4 T2I models, LMM rankings remain highly consistent (Spearman ≈ 0.98, Kendall ≈ 0.90). The average SIM variation across T2I models is only 0.013 (std).
> * Across 3 image encoders (CLIP, DINOv2, UNICOM), both SIM and FID show Spearman correlations of ≈ 0.98–0.99.
>
> These results indicate that auxiliary models introduce only minor noise and do not dominate the evaluation signal.

**Concern 2**: Human Study Design and Interpretability
> Human evaluation setup and consistency required clarification. **(Reviewer aasA, 8TQ1, iYF5)**

**Our Response:**
> We clarified the human study design and results (Appendix C.4):
> * The pipeline predictions were matched against all 9,000 individual votes, achieving 88.27% agreement, and 90.0% agreement with majority decisions.
> * Pattern-wise analysis shows stable agreement across diverse visual categories (87–92%), covering key skills such as object fidelity, symbolic understanding, and fine-grained detail.
>
> These results demonstrate that our human-study design is robust, interpretable, and provides reliable ground truth for evaluating pipeline alignment with human judgment.

**Concern 3**: Length Bias of LMM
>  The paper partially discusses instruction-following failure but does not experiment with length-controlled outputs. **(Reviewer aasA, 8TQ1, iYF5)**

**Our Response:**
> Length-controlled experiments (<20, 20–60, >60 words) show that (Appendix C.8):
> * Very short descriptions lack semantic coverage.
> * Longer descriptions yield moderate improvements.
> * Model rankings remain stable across all length ranges.
>
> This confirms that the benchmark rewards semantic completeness rather than verbosity.

**Concern 4**: Stochasticity and Reproducibility
> Evaluation results may be sensitive to randomness or single-run settings. **(Reviewer iYF5)**

**Our Response:**
> We performed a five-seed evaluation on 23 main models (Appendix C.9):
> * The std of SIM ranges from 0.0002 to 0.0038, while that of FID ranges from 0.0022 to 0.0141.
> * The relative ranking of the models remains highly consistent across different seeds, with Spearman = 0.9975 for SIM and Spearman = 0.9970 for FID.
>
> The reported scores closely match multi-seed means, demonstrating that stochastic effects are negligible.

---
We remain at your disposal for any further clarification or material. Thank you again for your time and effort in the review process.

---

### Meta-Review · Area_Chair_rYwP · 2025-12-09

**Summary:**

The reviewers gave a good comment of the contribution of this paper.
The reviewers mainly focused on three major weakness in this paper: 1) the impact of the embedded model and the T2I model on the evaluation results; 2) the length of the LLM's responses can lead to preference in the evaluation results, with longer instructions receiving higher scores; 3) the detailed setup of the human alignment experiment.

**Reviewer Concerns:**

Reviewer aasA and iYF5's questions have basically been fully answered.
For reviewer 8TQ1's question 4, the authors only provided a written explanation without adding relevant experimental results.
For reviewer 8ARY's question 3, the authors' answer did not provide a detailed description of the characteristics of GTP-4o.

**Reviewer Scores:**

Reviewer aasA, iYF5, and 8TQ1 may tend to increase the score, while reviewer 8ARY may tend to keep the score unchanged.

---

### Decision · Program_Chairs · 2026-01-26

Reject